# Calibrated Multimodal Representation Learning with Missing Modalities

**Xiaohao Liu**[1]  **Xiaobo Xia**[2]  **Jiaheng Wei**[3]  **Shuo Yang**[4]  **Xiu Su**[5]  **See-Kiong Ng**[1]  **Tat-Seng Chua**[1]

## Abstract

Multimodal representation learning harmonizes distinct modalities by aligning them into a unified latent space. Recent research generalizes traditional cross-modal alignment to produce enhanced multimodal synergy but requires all modalities to be present for a common instance, making it challenging to utilize prevalent datasets with missing modalities. We provide theoretical insights into this issue from an *anchor shift* perspective. Observed modalities are aligned with a local anchor that deviates from the optimal one when all modalities are present, resulting in an inevitable shift. To address this, we propose CalMRL to calibrate incomplete alignments caused by missing modalities. CalMRL leverages the priors and the inherent connections among modalities to model the imputation for the missing ones at the representation level. To resolve the optimization dilemma, we employ a bi-step learning method with the closed-form solution of the posterior distribution of shared latents. We validate its mitigation of anchor shift and convergence with theoretical guidance. By equipping the calibrated alignment with the existing advanced method, we offer new flexibility to absorb data with missing modalities, which is originally unattainable. Extensive experiments demonstrate the superiority of CalMRL. The code is released at `https://github.com/Xiaohao-Liu/CalMRL`.

## 1. Introduction

The empirical world is perceived by humans through diverse modalities, *e.g.*, vision, audio, and text (Ngiam et al., 2011;

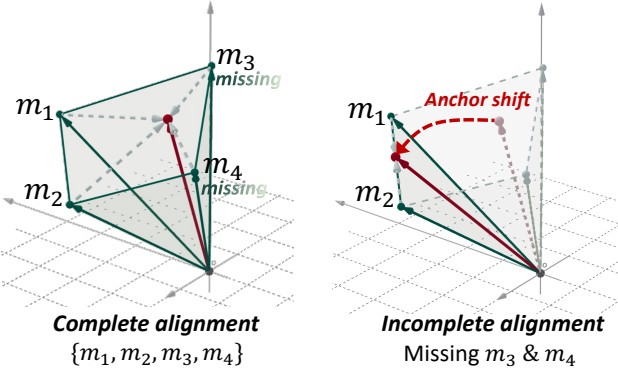

*Figure 1.* **Missing modalities result in distorted representation alignment.** Different modalities (in green) are aligned together with a virtual anchor (in red) implicitly with all modalities present. With missing modalities, observed ones are enforced to be aligned with a local anchor, deviating from the correct, *i.e.*, anchor shift.

Lu, 2023; Xu et al., 2023; Cao et al., 2025; Liang et al., 2021; Xin et al., 2025; Lei et al., 2024a; Gao et al., 2026; Lv et al., 2026). For example, eyes capture the shape or colors of one real instance, ears detect, and human language records. Different reflections on the instance shape its abstract form (Huh et al., 2024; Tjandrasuwita et al., 2025), which connects heterogeneous modalities. The single sound you heard invokes your rough imagination in the brain. Multimodal representation learning (Radford et al., 2021; Girdhar et al., 2023; Chen et al., 2023b; Cicchetti et al., 2025b; Dufumier et al., 2025; Liu et al., 2026), a key topic in multimodal learning, leverages aligned data to harmonize distinct unimodal encoders and bridges modalities, operating under this philosophy.

Recent research learns multiple modalities simultaneously, rather than grounding one to another (Radford et al., 2021; Wu et al., 2022; Xu et al., 2021; Luo et al., 2022; Guzhov et al., 2022; Zhang et al., 2022b), fostering greater synergy among them (Cicchetti et al., 2025b;a; Liu et al., 2026). To achieve multimodal alignment, they utilize geometric techniques to pull different unimodal representations together, ideally achieving a similar point (*i.e.*, anchor). Unfortunately, it requires training on datasets with all modalities present, *i.e.*, a complete set of modalities for one instance, thus ensuring an unbiased alignment. This introduces an inevitable challenge: collecting such comprehensive datasets is costly and contrasts with the prevalence of incomplete

[1]National University of Singapore [2]University of Science and Technology of China [3]The Hong Kong University of Science and Technology (Guangzhou) [4]Harbin Institute of Technology (Shenzhen) [5]Central South University. Correspondence to: Xiaobo Xia <xiaoboxia@ustc.edu.cn>.

*Proceedings of the 43rd International Conference on Machine Learning*, Seoul, South Korea. PMLR 306, 2026. Copyright 2026 by the author(s).

modality data, like paired data (*e.g.*, vision-text (Deng et al., 2009; Wang et al., 2024a; Nan et al., 2025) or audio-text (Kim et al., 2019; Drossos et al., 2020; Wu et al., 2023)). The disparity exists between different data, hindering a collective effect. In this paper, we frame this as a *missing modality* problem that is underexplored in multimodal representation learning.

Prior studies, like ImageBind (Girdhar et al., 2023) and LangugeBind (Zhu et al., 2023), integrate several paired data (*i.e.*, incomplete modality data), using one existing modality (*i.e.*, vision or text) as the aligning anchor (Guo et al., 2023; Chen et al., 2023b; Lyu et al., 2024; Wang et al., 2024c)[1]. To ensure the learning stability, the parameters of the corresponding encoder are fixed. They achieve aggregation of diverse modalities, yet bottlenecking the mutual improvements, and heavily depending on the performance of the fixed anchor. The issue of missing modalities is normally bypassed and set aside.

We delve into addressing this and analyzing with a natural perspective, anchor shifting. As illustrated in Figure 1, for complete alignment when all modalities are present, unimodal representations converge toward a virtual anchor within the space spanned by all modalities. In cases of incomplete alignment (*i.e.*, missing modalities $m_3$ and $m_4$), observed modalities are aligned with a local anchor, deviating from the optimal one associated with the instance, *i.e.*, resulting in *anchor shifting*. Reflecting on human perception, although we cannot touch or see a specific instance, we can still roughly connect it to other modalities via compensating for the missing ones, rather than being confined to solely what we can perceive. The compensation capability of humans inspires us to leverage the priors to impute missing modalities, thus calibrating the virtual anchor.

To this end, we introduce a novel multimodal representation learning framework, named **CalMRL**, which calibrates alignment in the presence of missing modalities. Specifically, we focus on the missing modality problem in the representation learning phase and theoretically analyze the inevitability of anchor shift. The main idea comes to light: construct modalities if they are missing, thus compensating for the anchor shift. We leverage the priors of missing modalities and the inherent connections among modalities, and introduce a simple generative model, where modalities share common latents, and meanwhile, possess their own distinctness. To achieve this, we adopt a two-step learning method. We first derive a closed-form solution for the posterior distribution of the shared latents with fixed parameters. Then, using this posterior, we optimize the generative parameters. By iterating these two steps, we can progressively refine the parameters, updating only with the

observed modalities. Missing modalities can be imputed by the shared latents, derived from the observations, and their priors. We further provide theoretical evidence to support the minor difference between the real and the imputed, and the convergence of our method. This method enables learning a calibrated alignment. We concatenate the observed and imputed modal representations and jointly optimize them with the objective of PMRL (Liu et al., 2026), aligning all modalities simultaneously. To validate the efficacy and rationale of CalMRL, we conduct extensive experiments and empirical analysis by comparing with state-of-the-art (SOTA) methods. Before delving into details, we summarize our contributions as follows:

- We introduce Calibrated Multimodal Representation Learning (CalMRL) to address the overlooked missing modalities dilemma, which leads to anchor shift in our theoretical analysis.

- To calibrate the alignment, we propose using a generative model to impute missing modalities, thus compensating for anchor shift. We refine the imputation precision by iterating the posterior inference and parameter optimization with theoretical grounding, followed by optimizing encoders by incorporating both observed and imputed modalities.

- We conduct extensive experiments to demonstrate the superiority of CalMRL and provide strong empirical evidence supporting our design rationale. Comprehensive empirical studies and discussions are also provided.

## 2. Preliminaries

**Aligning all modalities.** Different modalities produce synergies, presenting inherent connections despite the heterogeneity. Such connections serve as the key assumption to align different modalities together to learn multimodal representations (Lu, 2023; Liang et al., 2023). Contrastive learning successfully optimizes the similarities of modality pairs (Radford et al., 2021; Girdhar et al., 2023). Recent work proposes manipulating the singular value of a GRAM matrix to align multimodalities simultaneously (Liu et al., 2026). Specifically, all the uni-modal representations are concatenated together, *i.e.*, $\mathbf{z} = [\mathbf{z}^{m_1}, \cdots, \mathbf{z}^{|\mathcal{M}|}]$. $\sigma_1$ denotes the largest singular values of the GRAM matrix $\mathbf{zz}^\top$. By maximizing the largest one $\sigma_1$ among others $\{\sigma_i\}_{>1}$, it excels pair-wise methods in alignment tasks, yet is limited by predefined modalities, hard to extend for arbitrary modalities, and can be enhanced with modality-missing datasets.

**Modality missing.** Given the observed multimodal features $\mathbf{X}^\Omega$, the latent/unified representation can be derived via $p(\widetilde{\mathbf{z}}|\mathbf{X}^\Omega) = p(\widetilde{\mathbf{z}}|\{\mathbf{x}^m\}_{m\in\Omega})$. $\Omega$ is the set of observed

modalities and $\Omega \subseteq \mathcal{M}$, where $|\mathcal{M}| = k$. If $|\Omega| < k$, it incurs the phenomena of *modality missing*, which is quite common that some multimodal datasets only contain 2 modalities (Deng et al., 2009; Zhu et al., 2023), while some involve more (Chen et al., 2023b). For instance, ImageNet (Deng et al., 2009) only contains vision and text, while the VAST dataset (Chen et al., 2023b) additionally includes audio and subtitles. Our formulation is designed at the lower (instance) level to cover different cases, where we can view a dataset as a set of instances that miss the same modalities. That $\widetilde{\mathbf{z}}$ indicates the shared information across modalities. This further informs that the independence among modalities conditioned by $\widetilde{\mathbf{z}}$, formally, $\mathbf{x}^m \perp \mathbf{x}^{m'} | \widetilde{\mathbf{z}}, \forall \{m, m'\} \subset \mathcal{M}$.

**Probabilistic PCA.** It introduces a generative model which works as $\mathbf{x} = \mathbf{W}\mathbf{z} + \boldsymbol{\mu} + \boldsymbol{\epsilon}$, where $\boldsymbol{\mu} \in \mathbb{R}^{d'}$ and $\mathbf{z} \in \mathbb{R}^d$ follows $\mathcal{N}(\mathbf{0}, \mathbf{I})$ (Tipping & Bishop, 1999; Ghojogh et al., 2021). $\mathbf{W} \in \mathbb{R}^{d' \times d}$ transforms the latents $\mathbf{z}$ into observed space and $\boldsymbol{\epsilon} \sim \mathcal{N}(\mathbf{0}, \sigma^2 \mathbf{I})$. Therefore, the conditional distribution of the observed variable is $p(\mathbf{x}|\mathbf{z}) = \mathcal{N}(\mathbf{W}\mathbf{z} + \boldsymbol{\mu}, \sigma^2 \mathbf{I})$, and the marginal distribution is $p(\mathbf{x}) = \mathcal{N}(\boldsymbol{\mu}, \mathbf{W}\mathbf{W}^\top + \sigma^2 \mathbf{I})$. The closed-form solution is $\boldsymbol{\mu} = \frac{1}{N} \sum_{i=1}^{M} \mathbf{x}_n$, $\sigma^2 = \frac{1}{d'-d} \sum_{i=d+1}^{d'} \lambda_i$, and $\mathbf{W} = \mathbf{U}_d (\Lambda - \sigma^2 \mathbf{I})^{1/2}$. $\mathbf{U}\Lambda\mathbf{U}^\top = \mathrm{SVD}(\mathbf{S})$ decomposes the data covariance $\mathbf{S}$ to eigenvectors $\mathbf{U}$ and eigenvalue matrix $\Lambda = \mathrm{diag}(\lambda_1, \ldots, \lambda_{d'})$. We use similar modeling principles, specializing it for multimodal scenarios and emphasizing the connections and distinctiveness of modalities.

# 3. Calibrated Multimodal Representation Learning

In this section, we start with the formulation of the incomplete alignment, where the instance misses some modalities (*e.g.*, a text-audio pair missing its visual component), which also leads to the phenomenon of anchor shift. The absence of a modality causes the alignment center to deviate from the optimal "complete" multimodal space. We systematically analyze this problem and propose CalMRL by introducing an imputation mechanism that imputes representations for missing modalities in a latent space, effectively calibrating the alignment. We also provide a rigorous theoretical analysis of the alleviating effects and convergence analysis for the overall optimization objective.

**Incomplete alignment.** We consider a practical scenario where we cannot collect datasets with complete modalities. Formally, we represent the GRAM matrix as follows:

$$\mathbf{G}^{\Omega} = \begin{bmatrix} 1 & \langle \mathbf{z}^{m_1}, \mathbf{z}^{m_2} \rangle & \cdots & \langle \mathbf{z}^{m_1}, \mathbf{z}^{m_{k'}} \rangle \\ \langle \mathbf{z}^{m_2}, \mathbf{z}^{m_1} \rangle & 1 & \cdots & \langle \mathbf{z}^{m_2}, \mathbf{z}^{m_{k'}} \rangle \\ \vdots & \vdots & \ddots & \vdots \\ \langle \mathbf{z}^{m_{k'}}, \mathbf{z}^{m_1} \rangle & \langle \mathbf{z}^{m_{k'}}, \mathbf{z}^{m_2} \rangle & \cdots & 1 \end{bmatrix}$$

$$= \phi(\mathbf{X}^{\Omega})\phi(\mathbf{X}^{\Omega})^\top, \tag{1}$$

where $\mathbf{G}^{\Omega}$ is one of the block matrices belonging to the complete one $\mathbf{G} \in \mathbb{R}^{k \times k}$. In practice, when training on a mix of data, it is hard to ensure that all the data instances have the same modalities involved. In other words, one contains vision and text, while another might contain audio and text data. They all miss at least one modality of data to implement the complete alignment.

**Anchor shift $\boldsymbol{\Delta}(\mathbf{Z}, \mathbf{Z}^{\Omega})$.** Multiple modalities are aligned together with a virtual centroid (*i.e.*, anchor). Intuitively, there is a deviation between anchors in incomplete modalities and the complete ones. To confirm this, we also provide the theoretical evidence for the anchor shift.

**Theorem 1** (Anchor shift under incomplete modality alignment). *Let $\mathbf{u}_1$ and $\mathbf{u}_1^{\Omega}$ be the leading left singular vectors of the full multimodal matrix $\mathbf{Z}$ and its observed submatrix $\mathbf{Z}^{\Omega}$, respectively. Define $\sigma_1 = \|\mathbf{Z}\|_2$, $\sigma_1^{\Omega} = \|\mathbf{Z}^{\Omega}\|_2$, and $\eta := \sqrt{\sum_{m \in \bar{\Omega}} \langle \mathbf{u}_1^{\Omega}, \mathbf{z}^m \rangle^2}$. Then the anchor shift satisfies*

$$\sqrt{2\left(1 - \frac{\sigma_1^{\Omega} + \eta^2}{\sigma_1}\right)} \leq \underbrace{\|\mathbf{u}_1 - \mathbf{u}_1^{\Omega}\|}_{\|\boldsymbol{\Delta}\|} \leq \frac{\sqrt{2}\|\mathbf{Z}^{\bar{\Omega}}\|_2}{\sigma_1 - \sigma_2}.$$

**Remark.** The lower bound is *strictly positive* whenever the missing modalities contribute non-zero alignment ($\eta > 0$) or the observed data fails to capture the full leading singular value ($\sigma_1^{\Omega} < \sigma_1$). This implies that *any modality loss inevitably perturbs the virtual anchor*, no matter how well the remaining modalities are aligned. The upper bound shows that this perturbation can be mitigated, but never eliminated, by a large spectral gap (*i.e., strong alignment*) and limited energy in the missing modalities.

**Missing modalities via imputation.** We design the generation of unimodal representation to maintain intermodality connections following the conventional principles (Lu, 2023; Gupta et al., 2025):

$$\mathbf{z}^m = \mathbf{W}^m \boldsymbol{\beta} + \boldsymbol{\mu}^m + \boldsymbol{\epsilon}^m, \boldsymbol{\epsilon}^m \sim \mathcal{N}(\mathbf{0}, (\sigma^m)^2 \mathbf{I}). \tag{2}$$

The representation for modality $m$ is conditioned by shared latents $\boldsymbol{\beta}$ and its uniqueness $\boldsymbol{\mu}_m$. In this case, we can impute the missing modalities at the representation level to compensate for the anchor offset, leading to $\boldsymbol{\Delta}(\mathbf{Z}, [\mathbf{Z}^{\Omega}; \mathbf{Z}^{\bar{\Omega}}]) < \boldsymbol{\Delta}(\mathbf{Z}, \mathbf{Z}^{\Omega})$. To this end, even with the missing modalities, the observed modalities can be aligned in an approximated environment with complete modalities.

To optimize the parameters, we aim to maximize the marginal log-likelihood:

$$\arg\max_{\widehat{\boldsymbol{\theta}}} \sum_{i=1}^{N} \log p(\mathbf{z}) = \arg\max_{\widehat{\boldsymbol{\theta}}} \sum_{i=1}^{N} \log \int p(\mathbf{z} \mid \boldsymbol{\beta}) p(\boldsymbol{\beta}) \, d\boldsymbol{\beta}, \tag{3}$$

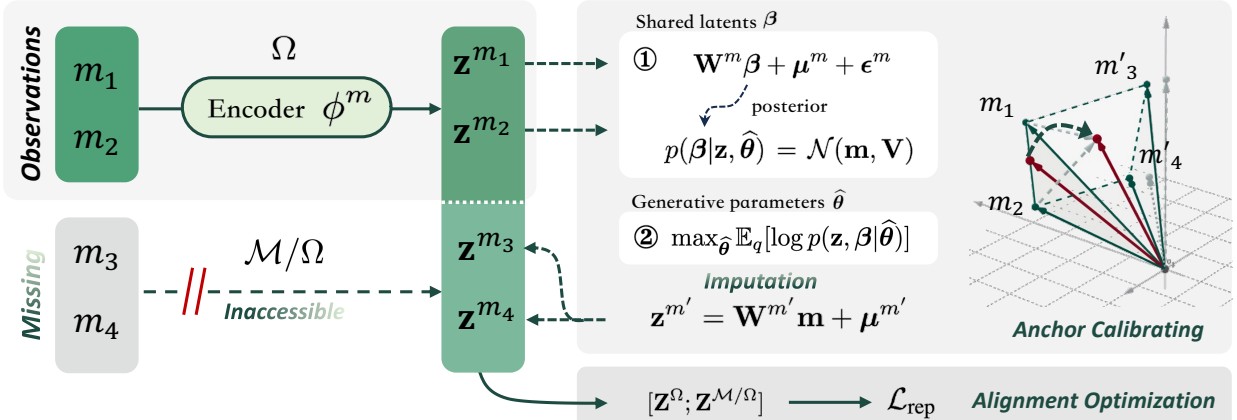

*Figure 2.* **The overall framework of CalMRL.** Observed unimodal content is first encoded to corresponding representations $\{\mathbf{z}^m\}_{m\in\Omega}$ with individual encoders $\phi^m$ in $\boldsymbol{\theta}$. Despite the missing modalities (*i.e.*, $\mathcal{M}/\Omega$), CalMRL calibrates multimodal alignment whereby missing modalities are imputed by generative parameters $\widehat{\boldsymbol{\theta}}$. Finally, $\mathcal{L}_{\text{rep}}$ optimizes the observed unimodal encoder to be aligned with the calibrated direction.

where $N$ denotes the batch size and $\widehat{\boldsymbol{\theta}} = \{\mathbf{W}^m, \boldsymbol{\mu}^m, \sigma^m\}_{m\in\mathcal{M}}$. Unfortunately, as different unimodal parameters are hinged with the common $\boldsymbol{\beta}$, we cannot resolve the parameters with a closed-form solution via naive Probabilistic PCA. For each example, we introduce an arbitrary distribution $q(\boldsymbol{\beta})$ and derive the evidence lower bound:

$$\log p(\mathbf{z}) = \log \mathbb{E}_q\left[\frac{p(\mathbf{z},\boldsymbol{\beta})}{q(\boldsymbol{\beta})}\right] \geq \mathbb{E}_q \log\left[\frac{p(\mathbf{z},\boldsymbol{\beta})}{q(\boldsymbol{\beta})}\right]. \quad (4)$$

By iteratively optimizing $q$ and $\widehat{\boldsymbol{\theta}}$, we can elevate the likelihood.

**Resolving $\widehat{\boldsymbol{\theta}}$.** Accordingly, we obtain the generative parameters with a bi-step optimization.

① Fixing the parameters, we optimize the lower bound concerning $q$, *i.e.*, $\max_q \mathbb{E}_q[\log p(\mathbf{z},\boldsymbol{\beta}|\widehat{\boldsymbol{\theta}})] - \mathbb{E}_q[\log q(\boldsymbol{\beta})]$. The equality of Eq. (4) holds if and only if $q(\boldsymbol{\beta}) = p(\boldsymbol{\beta}|\mathbf{z},\widehat{\boldsymbol{\theta}})$. Besides, we can obtain the posterior $p(\boldsymbol{\beta}|\mathbf{z},\widehat{\boldsymbol{\theta}}) = \mathcal{N}(\mathbf{m},\mathbf{V})$ in a Gaussian distribution. In particular,

$$\begin{aligned} \mathbf{V} &= \left[\mathbf{I} + \sum_{m\in\Omega} \frac{1}{(\sigma^m)^2}\mathbf{W}^{m\top}\mathbf{W}^m\right]^{-1}, \\ \mathbf{m} &= \mathbf{V}\sum_{m\in\Omega}\frac{1}{(\sigma^m)^2}\mathbf{W}^{m\top}(\mathbf{z}^m - \boldsymbol{\mu}^m), \end{aligned} \quad (5)$$

and we can update *only* with the observed modalities in $\Omega$.

② Given the posterior, we can maximize the lower bound concerning $\widehat{\boldsymbol{\theta}}$, *i.e.*, $\max_{\widehat{\boldsymbol{\theta}}} \mathbb{E}_q[\log p(\mathbf{z},\boldsymbol{\beta}|\widehat{\boldsymbol{\theta}})]$. Here we derive the closed-form solution for each parameter as follows:

$$\begin{cases} \boldsymbol{\mu}^m = \dfrac{1}{N}\displaystyle\sum_{i=1}^N(\mathbf{z}_i^m - \mathbf{W}^m\mathbf{m}_i), \\[2mm] \mathbf{W}^m = \left(\displaystyle\sum_{i=1}^N(\mathbf{z}_i^m - \boldsymbol{\mu}^m)\mathbf{m}_i^\top\right)\left(\displaystyle\sum_{i=1}^N \mathbb{E}[\boldsymbol{\beta}_i\boldsymbol{\beta}_i^\top]\right)^{-1}, \\[2mm] (\sigma^m)^2 = \dfrac{1}{Nd}\displaystyle\sum_{i=1}^N\Big[\|\mathbf{z}_i^m - \boldsymbol{\mu}^m - \mathbf{W}^m\mathbf{m}_i\|^2 \\[2mm] \qquad\qquad + \operatorname{Tr}[\mathbf{W}^{m\top}\mathbf{W}^m\mathbf{V}]\Big]. \end{cases} \quad (6)$$

Alternatively, the parameters of the model can be optimized by minimizing the Gaussian negative log-likelihood loss. These generative parameters can be refined by the next iteration. For the generation of representation for missing modalities $\mathbf{z}^{m'}$, we have the following proposition.

**Proposition 2** (Missing modality imputation). *Given observed modalities $\Omega$ and learned generative parameters $\widehat{\boldsymbol{\theta}} = \{\mathbf{W}^m, \boldsymbol{\mu}^m, \sigma^m\}_{m\in\mathcal{M}}$, the representation of any missing modality $m' \in \mathcal{M}\setminus\Omega$ can be imputed in closed form as*

$$\widehat{\mathbf{z}}^{m'} = \mathbf{W}^{m'}\mathbf{m} + \boldsymbol{\mu}^{m'}, \quad \forall m' \in \mathcal{M}/\Omega. \quad (7)$$

We provide the detailed derivation in Appendix A.2.

**Training objective.** For the next phase, we optimize the encoders $\phi : \mathcal{X} \to \mathcal{Z}$ with parameters $\boldsymbol{\theta}$. We complete the multimodal representations and execute the maximum singular values following (Liu et al., 2026).

$$\mathbf{U}\boldsymbol{\Lambda}\mathbf{V} = \text{SVD}([\mathbf{Z}^\Omega; \widehat{\mathbf{Z}}^{\mathcal{M}/\Omega}]), \quad \boldsymbol{\Lambda} = \operatorname{diag}(\lambda_1,\dots,\lambda_k). \quad (8)$$

Accordingly, the learning objective is designed to maximize the largest singular value to enforce full alignment, and the corresponding eigenvectors are used for instance-level regularization, ensuring uniformity, as follows:

$$\mathcal{L}_{\text{rep}} = -\frac{1}{N}\sum_{i=1}^N\left[\frac{\exp[\lambda_1/\tau]}{\sum_{j=1}^k \exp[\lambda_j/\tau]} + \frac{\exp[\mathbf{u}_1^{i\top}\mathbf{u}_1^i/\tau']}{\sum_{j=1}^N \exp[\mathbf{u}_1^{i\top}\mathbf{u}_1^j/\tau']}\right]$$

$$+ \alpha \cdot \mathbb{E}_{\{m\}^k \sim \mathcal{M}}[y \log \hat{y} + (1 - y) \log(1 - \hat{y})], \quad (9)$$

where $N$ denotes the number of data points in a batch. The instance matching loss is applied only to observed modalities, and weighted by $\alpha = 0.1$ with a multimodal encoder and an MLP layer to predict whether the multimodal data is matched or not, returning the prediction $\hat{y}$ (Chen et al., 2023b; Liu et al., 2026; Cicchetti et al., 2025a).

### 3.1. Analysis

To shed light on our method, we provide a more in-depth analysis of how it alleviates the anchor shift and whether it can converge.

**Alleviating anchor shift.** CalMRL introduces the imputation of missing modalities to mitigate the anchor shift in multimodal alignment. We derive the comparison results of the anchor offset before and after calibration as follows.

**Corollary 3** (Less anchor shift with calibration). *Let each imputation satisfies* $\|\widehat{\mathbf{z}}^{m'} - \mathbf{z}^{m'}\|_2 \leq \varepsilon, \forall m' \in \bar{\Omega}$. *Then the calibration-induced anchor deviation obeys* $\|\boldsymbol{\Delta}(\mathbf{Z}, [\mathbf{Z}^\Omega; \mathbf{Z}^{\bar{\Omega}}])\| < \|\boldsymbol{\Delta}(\mathbf{Z}, \mathbf{Z}^\Omega)\|$ *if and only if*

$$\varepsilon < \frac{\sigma_1 - \sigma_2}{\sqrt{|\bar{\Omega}|}} \sqrt{1 - \frac{\sigma_1^\Omega + \eta^2}{\sigma_1}}. \quad (10)$$

**Remark.** This corollary reveals that the benefit of calibration for anchor shift, especially in the case of strong alignment among modalities ($\uparrow \sigma_1 - \sigma_2$) and relatively small contribution of missing modalities ($\downarrow \frac{\sigma_1^\Omega + \eta^2}{\sigma_1}$).

**The convergence analysis.** CalMRL utilizes the bi-step iteration to optimize the generative parameters, which raises concerns about its convergence. To answer this, we confirm the monotonicity of the log-likelihood as follows.

**Corollary 4** (Monotonicity for CalMRL imputation). *Let* $\widehat{\boldsymbol{\theta}}^{(t)}$ *be the parameters at iteration* $t$ *applied to the observed-data log-likelihood* $L(\widehat{\boldsymbol{\theta}}) = \sum_n \log p(\mathbf{z}_n^\Omega \mid \widehat{\boldsymbol{\theta}})$ *under the generative model in Eq. (2). Given the solution of* $q$ *and* $\widehat{\boldsymbol{\theta}}$, $L(\widehat{\boldsymbol{\theta}}^{(t+1)}) \geq L(\widehat{\boldsymbol{\theta}}^{(t)})$.

**Remark.** The monotonic increase in likelihood guarantees stable convergence of the generative model, generally to a local stationary point (see proof in Appendix A.4). In CalMRL, this subroutine operates within a larger alternating-optimization framework that updates the unimodal encoders, thereby providing a solid foundation for the overall convergence of the full training procedure.

### 3.2. Training Algorithm Flow

To implement and clarify our method, we provide the algorithm flow for training in Algorithm 1. Building upon

---

**Algorithm 1** CalMRL (Training)

**Require:** Multimodal dataset $\{\mathbf{x}_i\}_{i=1}^N$; encoder parameters $\boldsymbol{\theta}$; generative parameters $\widehat{\boldsymbol{\theta}}$; $\tau, \tau'$; $\alpha$
**Ensure:** $\phi_{\boldsymbol{\theta}}$ and $\widehat{\boldsymbol{\theta}}$
1: Initialize parameters $\phi_{\boldsymbol{\theta}}$ and $\widehat{\boldsymbol{\theta}}$
2: **for** each batch $\mathcal{B} = \{\mathbf{x}_i\}_{i=1}^N$ **do**
3:     Encode: $\mathbf{z}_i^m = \phi_{\boldsymbol{\theta}}^m(\mathbf{x}_i^m), \forall m \in \Omega_i$

> **Missing modalities via imputation**
>
> Update: $\begin{cases} \mathbf{m}_i, \mathbf{V}_i \leftarrow \text{Eq. (5)} \\ \widehat{\boldsymbol{\theta}} = \{\mathbf{W}^m, \boldsymbol{\mu}^m, \sigma^m\}_\Omega \leftarrow \text{Eq. (6)} \end{cases}$
>
> Impute: $\mathbf{z}_i^{m'} \leftarrow \mathbf{W}^{m'}\mathbf{m}_i + \boldsymbol{\mu}^{m'}, m' \in \mathcal{M} \setminus \Omega_i$

4:
5:     Update $\boldsymbol{\theta}$: $\boldsymbol{\theta} \leftarrow \mathcal{L}_{\text{rep}}$ (Eq. (9))
6: **end for**

---

our theoretical derivation, our method is straightforward, applying a few steps to update the parameters in batches of data. Observed unimodal content will be encoded and then used to determine the posterior, which also makes the generative parameters resolvable within this multimodal learning framework.

## 4. Experiments

In this section, we conduct experiments to address the following research question:

- **RQ1:** Does CalMRL outperform other multimodal representation methods under the missing-modality setting?

- **RQ2:** What is the contribution of CalMRL to different missing modalities? Does it satisfy the theoretical insight to calibrate for a better alignment with empirical results?

- **RQ3:** How is the training of CalMRL stable? Can it maintain the distribution of multimodal representations?

### 4.1. Experimental Setup

**Datasets.** We first adopt the VAST-150K with complete modalities for warming up parameters, following (Chen et al., 2023b; Cicchetti et al., 2025b). Then we train the model on datasets with missing modality. Specifically, we select two datasets: MSRVTT (Bain et al., 2021) for vision-text pairs and AudioCaps (Kim et al., 2019) for audio-text pairs, covering two mainstream types of multimodal datasets (*i.e.*, V↔T, A↔T). To evaluate our method, we select 10 benchmarks, including MSR-VTT (Chen & Dolan, 2011), DiDeMo (Anne Hendricks et al., 2017), ActivityNet (Krishna et al., 2017), and VATEX (Wang et al., 2019) for vision-text retrieval, AudioCaps (Kim et al., 2019), and

*Table 1.* **Multimodal retrieval results (%)** in terms of Recall@1 on video-text (T→V and V→T) and audio-text (T→A and A→T) datasets. "↑" indicates the model continually trained with missing modality datasets. The best result in each case is marked in bold, and the second-best result is underlined. Increment points are computed compared with VAST.

| | MSR-VTT | | DiDeMo | | ActivityNet | | VATEX | | AudioCaps | | Clotho | | Avg. |
|---|---|---|---|---|---|---|---|---|---|---|---|---|---|
| | T→V | V→T | T→V | V→T | T→V | V→T | T→V | V→T | T→A | A→T | T→A | A→T | |
| ImageBind | 36.8 | - | - | - | - | - | - | - | 9.3 | - | 6.0 | - | - |
| InternVideo-L | 40.7 | 39.6 | 31.5 | 33.5 | 30.7 | 31.4 | 49.5 | 69.5 | - | - | - | - | - |
| LanguageBind | 44.8 | 40.9 | 39.9 | 39.8 | 41.0 | 39.1 | - | - | 19.7 | - | 16.7 | - | - |
| VAST | 50.5 | 49.0 | 48.6 | 46.9 | 51.7 | 48.8 | 75.9 | 74.8 | 33.7 | 32.2 | 12.4 | 13.0 | 44.8 |
| GRAM | $52.1^{+1.6}$ | $51.8^{+2.8}$ | $53.1^{+4.5}$ | $50.7^{+3.8}$ | $54.5^{+2.8}$ | $48.3^{-0.5}$ | $77.5^{+1.6}$ | $74.7^{-0.1}$ | $34.6^{+0.9}$ | $35.2^{+3.0}$ | $15.9^{+3.5}$ | $16.2^{+3.2}$ | 47.1 |
| TRIANGLE | $54.3^{+3.8}$ | $51.7^{+2.7}$ | $53.4^{+4.8}$ | $52.7^{+5.8}$ | $55.4^{+3.7}$ | $50.9^{+2.1}$ | $79.9^{+4.0}$ | $74.8^{+0.0}$ | $37.2^{+3.5}$ | $37.2^{+5.0}$ | $15.3^{+2.9}$ | $13.7^{+0.7}$ | 48.0 |
| PMRL | $55.1^{+4.6}$ | $53.5^{+4.5}$ | $53.5^{+4.9}$ | $51.3^{+4.4}$ | $56.0^{+4.3}$ | $49.6^{+0.8}$ | $80.5^{+4.6}$ | $75.2^{+0.4}$ | $36.1^{+2.4}$ | $33.9^{+1.7}$ | $16.8^{+4.4}$ | $16.1^{+3.1}$ | 48.1 |
| VAST↑ | $58.5^{+8.0}$ | $\underline{60.2}^{+11.2}$ | $53.9^{+5.3}$ | $53.1^{+6.2}$ | $55.7^{+4.0}$ | $\underline{53.9}^{+5.1}$ | $80.0^{+4.1}$ | $77.9^{+3.1}$ | $49.1^{+15.4}$ | $\mathbf{53.3}^{+21.1}$ | $21.8^{+9.4}$ | $21.8^{+8.8}$ | 53.3 |
| GRAM↑ | $59.7^{+9.2}$ | $57.2^{+8.2}$ | $54.8^{+6.2}$ | $53.1^{+6.2}$ | $\underline{56.2}^{+4.5}$ | $53.5^{+4.7}$ | $\underline{80.5}^{+4.6}$ | $79.2^{+4.4}$ | $49.1^{+15.4}$ | $51.7^{+19.5}$ | $20.6^{+8.2}$ | $19.5^{+6.5}$ | 52.9 |
| TRIANGLE↑ | $57.6^{+7.1}$ | $58.4^{+9.4}$ | $51.7^{+3.1}$ | $51.1^{+4.2}$ | $54.2^{+2.5}$ | $51.0^{+2.2}$ | $77.9^{+2.0}$ | $76.6^{+1.8}$ | $48.3^{+14.6}$ | $51.7^{+19.5}$ | $19.9^{+7.5}$ | $20.2^{+7.2}$ | 51.6 |
| PMRL↑ | $\underline{60.1}^{+9.6}$ | $59.2^{+10.2}$ | $\underline{55.1}^{+6.5}$ | $\underline{53.3}^{+6.4}$ | $55.8^{+4.1}$ | $\mathbf{54.0}^{+5.2}$ | $80.4^{+4.5}$ | $\underline{78.7}^{+3.9}$ | $\mathbf{50.4}^{+16.7}$ | $52.0^{+19.8}$ | $\mathbf{23.5}^{+11.1}$ | $\mathbf{23.1}^{+10.1}$ | $\underline{53.8}$ |
| **CalMRL↑** | $\mathbf{61.1}^{+10.6}$ | $\mathbf{61.1}^{+12.1}$ | $\mathbf{55.4}^{+6.8}$ | $\mathbf{53.7}^{+6.8}$ | $\mathbf{57.1}^{+5.4}$ | $53.6^{+4.8}$ | $\mathbf{81.3}^{+5.4}$ | $\mathbf{79.2}^{+4.4}$ | $\underline{50.1}^{+16.4}$ | $51.0^{+18.8}$ | $\mathbf{23.8}^{+11.4}$ | $\underline{22.4}^{+9.4}$ | **54.2** |

*Table 2.* **Multimodal classification results (%)** in terms of Accuracy on video-text and audio-text datasets across different models.

| | VGGSound | UCF101 | AudioSet | ESC50 | Avg. |
|---|---|---|---|---|---|
| ImageBind | 25.34 | 69.31 | 14.92 | 58.74 | 42.08 |
| VAST↑ | 25.98 | 74.55 | 15.03 | 59.99 | 43.89 |
| GRAM↑ | 25.78 | 73.56 | 15.65 | 58.50 | 43.37 |
| TRIANGLE↑ | 25.69 | 73.86 | 15.20 | 59.24 | 43.50 |
| PMRL↑ | 25.24 | 77.13 | 15.29 | 58.50 | 44.04 |
| **CalMRL↑** | $\mathbf{26.09}^{+0.85}$ | $\mathbf{78.91}^{+1.78}$ | $\mathbf{16.01}^{+0.72}$ | $\mathbf{59.75}^{+1.25}$ | $\mathbf{45.19}^{+1.15}$ |

Clotho (Drossos et al., 2020) for audio-text retrieval. Only test splits are used for evaluation to avoid any information leakage. We also evaluate our method on the multimodal classification task via fine-tuning on VGGSound (Chen et al., 2020a) and UCF101 (Soomro et al., 2012), and evaluating on four testing datasets, including VGGSound, UCF101, AudioSet (Gemmeke et al., 2017), and ESC50 (Piczak, 2015), in Section 4.2. We detail the statistics of datasets in Appendix C.1.

**Baselines & evaluation metrics.** We compare our method with existing well-trained models, including ImageBind (Girdhar et al., 2023), InternVideo (Wang et al., 2022), LanguageBind (Zhu et al., 2023), VAST (Chen et al., 2023b), GRAM (Cicchetti et al., 2025b), TRIANGLE (Cicchetti et al., 2025a), and PMRL (Liu et al., 2026). We detail the implementation of baselines, especially their adaptation for missing modalities in Appendix C.3. Performance is evaluated using Recall@1 for retrieval and Accuracy for classification. Detailed results with different top-k ($\{5, 10\}$) on retrieval tasks are shown in Appendix D, with results reported for bidirectional retrieval (*e.g.*, T→V and V→T).

**Implementation details.** We implement our model upon VAST (Chen et al., 2023b) which supports four modalities, *i.e.*, vision, caption, audio, and subtitle. We also efficiently adapt different multimodal alignment methods on Image-Bind evaluate its classification performance. The detailed model architecture is in Appendix C.2. VAST, GRAM, TRI-

ANGLE, and PMRL are all fine-tuned on the same training datasets aligning with the missing modality training to ensure a fair comparison with CalMRL. We also report their base performance only trained on the full modality dataset (VAST) in our empirical results (*cf.*, Section 4.3).

## 4.2. Performance Comparison (RQ1)

We report the results on ten datasets, covering two main multimodal retrieval and classification tasks, as shown in Tables 1 and 2. All the models are trained on a complete-modality dataset initially, and "↑" denotes the continual training on missing modality datasets. Accordingly, we draw the following conclusions:

① *Learning with complete-modality simultaneously achieves higher performance.* Initial models, *e.g.*, Image-Bind, InternVideo-L, and LanguageBind, extend the pairwise contrastive learning paradigm. However, they generally perform worse than models trained on datasets where all modalities are available simultaneously. New learning objectives have been introduced specifically to optimize multiple modalities concurrently. Among these methods, PMRL demonstrates relatively better performance. The aligned modalities naturally reflect different aspects of a single common instance. This inherent alignment intuitively provides significant benefits for multimodal representation learning.

② *Missing modalities can boost the model yet incrementally.* When certain modalities are missing, virtually all models demonstrate performance improvements, particularly on in-domain datasets (*i.e.*, MSR-VTT and AudioCaps). Observable performance gains are also generalized to various out-of-domain datasets, such as DiDeMo and ActivityNet, though these gains are incremental. The performance boost on audio-text datasets (*i.e.*, AudioCaps and Clotho) is sig-

*Table 3.* **Multimodal retrieval results (%) for models trained on sole datasets.** "↑$^{VT}$" and "↑$^{AT}$" indicate the model continually trained with video-text and audio-text modality datasets, respectively. The best result in each case is marked in bold, and the second-best result is underlined. Increment points are computed compared with VAST.

| | MSR-VTT | | DiDeMo | | ActivityNet | | VATEX | | AudioCaps | | Clotho | | Avg. |
|---|---|---|---|---|---|---|---|---|---|---|---|---|---|
| | T→V | V→T | T→V | V→T | T→V | V→T | T→V | V→T | T→A | A→T | T→A | A→T | |
| VAST↑$^{AT}$ | $53.1^{+2.6}$ | $50.9^{+1.9}$ | $45.0^{-3.6}$ | $46.0^{-0.9}$ | $48.8^{-2.9}$ | $48.5^{-0.3}$ | $77.3^{+1.4}$ | $75.9^{+1.1}$ | $51.1^{+17.4}$ | $52.1^{+19.9}$ | $21.3^{+8.9}$ | $21.1^{+8.1}$ | 49.3 |
| GRAM↑$^{AT}$ | $49.0^{-1.5}$ | $49.3^{+0.3}$ | $48.5^{-0.1}$ | $48.3^{+1.4}$ | $49.2^{-2.5}$ | $48.0^{-0.8}$ | $58.1^{-17.8}$ | $74.1^{-0.7}$ | $\underline{53.0}^{+19.3}$ | $52.1^{+19.9}$ | $\underline{22.6}^{+10.2}$ | $\mathbf{22.5}^{+9.5}$ | 47.9 |
| TRIANGLE↑$^{AT}$ | $55.8^{+5.3}$ | $52.4^{+3.4}$ | $\underline{50.1}^{+1.5}$ | $48.9^{+2.0}$ | $50.2^{-1.5}$ | $\mathbf{50.0}^{+1.2}$ | $\mathbf{79.8}^{+3.9}$ | $76.1^{+1.3}$ | $47.0^{-13.3}$ | $50.9^{+18.7}$ | $16.6^{+4.2}$ | $18.4^{+5.4}$ | 49.7 |
| PMRL↑$^{AT}$ | $\underline{56.2}^{+5.7}$ | $\underline{52.7}^{+3.7}$ | $48.7^{+0.1}$ | $\underline{49.4}^{+2.5}$ | $\underline{52.0}^{+0.3}$ | $49.3^{+0.5}$ | $78.8^{+2.9}$ | $\mathbf{76.6}^{+1.8}$ | $52.0^{+18.3}$ | $\underline{54.0}^{+21.8}$ | $\mathbf{22.7}^{+10.3}$ | $\underline{21.8}^{+8.8}$ | 51.2 |
| **CalMRL↑$^{AT}$** | $\mathbf{56.4}^{+5.9}$ | $\mathbf{53.3}^{+4.3}$ | $\mathbf{50.5}^{+1.9}$ | $\mathbf{50.7}^{+3.8}$ | $\mathbf{53.8}^{+2.1}$ | $\underline{49.8}^{+1.0}$ | $\underline{79.2}^{+3.3}$ | $\mathbf{76.6}^{+1.8}$ | $\mathbf{53.1}^{+19.4}$ | $\mathbf{54.1}^{+21.9}$ | $21.3^{+8.9}$ | $21.6^{+8.6}$ | **51.7** |
| VAST↑$^{VT}$ | $59.5^{+9.0}$ | $\mathbf{60.3}^{+11.3}$ | $54.6^{+6.0}$ | $\mathbf{55.1}^{+8.2}$ | $\mathbf{57.6}^{+5.9}$ | $\mathbf{54.9}^{+6.1}$ | $\underline{80.1}^{+4.2}$ | $78.3^{+3.5}$ | $31.5^{-2.2}$ | $33.4^{+1.2}$ | $15.5^{+3.1}$ | $15.5^{+2.5}$ | 49.7 |
| GRAM↑$^{VT}$ | $60.1^{+9.6}$ | $57.8^{+8.8}$ | $\underline{56.5}^{+7.9}$ | $54.0^{+7.1}$ | $56.2^{+4.5}$ | $\underline{53.5}^{+4.7}$ | $78.9^{+3.0}$ | $78.3^{+3.5}$ | $\mathbf{32.5}^{-1.2}$ | $\mathbf{36.8}^{+4.6}$ | $16.1^{+3.7}$ | $15.2^{+2.2}$ | 49.7 |
| TRIANGLE↑$^{VT}$ | $57.8^{+7.3}$ | $58.7^{+9.7}$ | $53.1^{+4.5}$ | $50.9^{+4.0}$ | $56.3^{+4.6}$ | $51.6^{+2.8}$ | $78.4^{+2.5}$ | $77.3^{+2.5}$ | $31.2^{-2.5}$ | $32.5^{+0.3}$ | $16.3^{+3.9}$ | $14.7^{+1.7}$ | 48.2 |
| PMRL↑$^{VT}$ | $\underline{60.7}^{+10.2}$ | $\underline{60.0}^{+11.0}$ | $56.1^{+7.5}$ | $53.5^{+6.6}$ | $\underline{57.5}^{+5.8}$ | $53.2^{+4.4}$ | $79.7^{+3.8}$ | $78.1^{+3.3}$ | $32.0^{-1.7}$ | $\underline{34.8}^{+2.6}$ | $\underline{18.2}^{+5.8}$ | $\mathbf{15.9}^{+2.9}$ | 50.0 |
| **CalMRL↑$^{VT}$** | $\mathbf{61.1}^{+10.6}$ | $\mathbf{60.3}^{+11.3}$ | $\mathbf{57.5}^{+8.9}$ | $\underline{54.4}^{+7.5}$ | $\underline{57.5}^{+5.8}$ | $52.2^{+3.4}$ | $\mathbf{81.5}^{+5.6}$ | $\mathbf{79.4}^{+4.6}$ | $31.4^{-2.3}$ | $\underline{34.8}^{+2.6}$ | $\mathbf{18.9}^{+6.5}$ | $\underline{15.8}^{-2.8}$ | **50.4** |

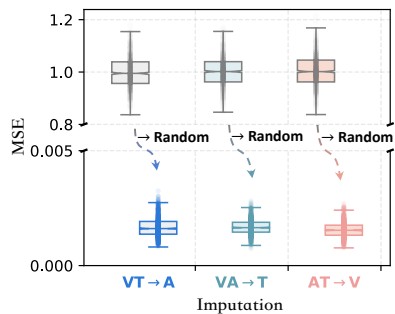

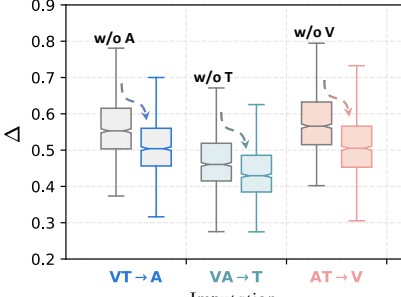

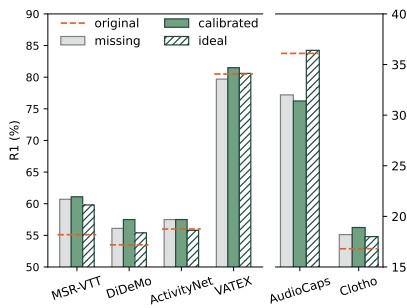

*Figure 3.* **MSEs between real and imputed representations.** "→" marks the direction of imputation; **Random** refers to representations drawn at random.

*Figure 4.* **Comparison of anchor shift (∆) before and after calibration.** The left box with a gray border shows the anchor shift with missing modalities (w/o).

*Figure 5.* **The performance comparison across missing, calibrated, and full ("ideal") modalities.** All the models are trained on MSR-VTT.

nificantly greater compared to the boost observed on vision-text alignment tasks. This disparity indicates that vision-text alignment has benefited from extensive prior research and advancements, whereas audio-text alignment appears to have substantially greater room for improvement (corresponds to the visualization in Figure 6).

③ *CalMRL further improves without incorporating new information.* Among all the methods, CaLMRL demonstrates superior performance improvements over its backbone model, VAST, and surpasses the SOTA methods in most cases. Rather than requiring new datasets, CaLMRL adapts existing bimodal datasets to enhance the model using a simple generative approach, showcasing strong promise.

④ *Further classification results also reflect similar effects.* In general, CaLMRL can also achieve consistent performance gains across diverse multimodal classification benchmarks, including UCF101, VGGSound, AudioSet, and ESC50 tasks. As shown in Table 2, CaLMRL outperforms the baselines, reaching an average accuracy of 45.19%. Notably, it achieves a significant improvement of +1.78% on UCF101 tuning with the same strategy as PMRL. These results further validate that the representation alignment

and generative imputation in CaLMRL effectively enhance the model's robustness and generalization capabilities in missing-modality scenarios.

### 4.3. Further Analysis (R2 & R3)

**Sole datasets with calibration (R2).** To unravel the effect of different missing modality datasets, we conduct experiments on a single dataset with calibration. Table 3 showcases the performance evaluated on different variants of models. "↑$^{AT}$" indicates the training with only audio-text dataset (*i.e.*, AudioCaps). The results for all given benchmarks are reported, and we have the following insights. ① *Training on the sole dataset can improve the relevant capabilities of modalities while harming others.* For instance, training on an audio-text dataset, we can observe a significant improvement on AudioCpas and Clotho (out-of-domain), and a performance drop on other video-text datasets, and vice versa. Such harmfulness can be found clearly under only the vision-text dataset training. ② *CalMRL heals to some extent and even sometimes achieves better performance.* In audio-text training, CalMRL resists performance degradation in most vision-text benchmarks, and even gains better results in AudioCaps. Combining the

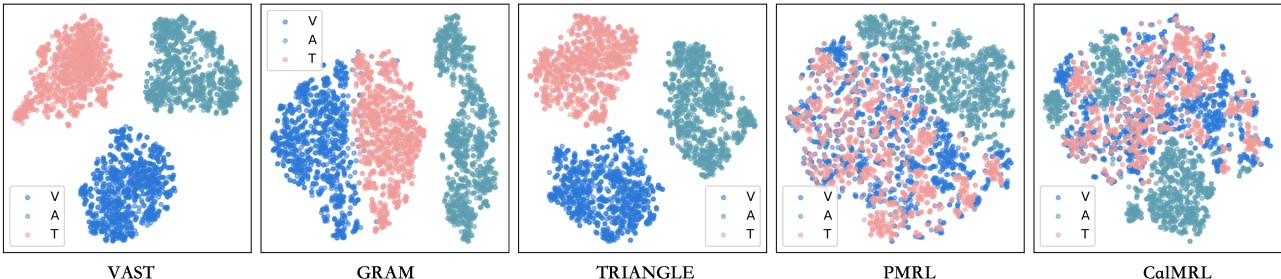

*Figure 6.* **t-SNE visualization on multimodal representations generated by different models.** Existing models, under missing modality training, present clearly separated clusters for each modality (distinct modal boundaries). Fortunately, CalMRL mitigates this issue.

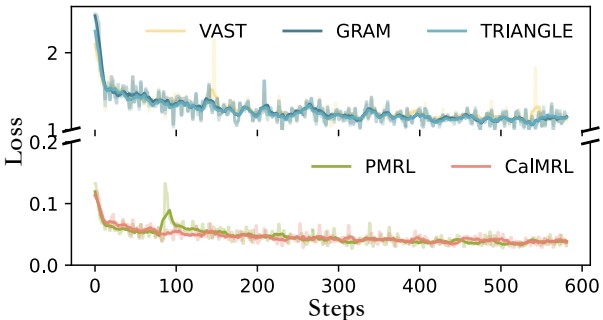

*Figure 7.* Loss curves across models on the training phase.

results in Table 1, CalMRL demonstrates the collective benefit of training on a mixture of datasets, resulting in improved performance. These results provide strong evidence to shed light on how CalMRL works: calibrating the alignment to resist the degradation for missing modalities to maintain an overall better performance.

**Compensating for the anchor shift (R2).** We measure the anchor shift before and after calibration to validate its effectiveness. Figures 3 and 4 illustrate that ① *our method can effectively reconstruct the original representations* and ② *the anchor shift is mitigated as expected*. This indicates that our method, even with a simple generative model, is capable of learning multimodal connections at the representation level. Without introducing any new information, the incomplete alignment can be approximated to the complete one via the mitigated anchor shift.

**Approaching to the *"ideal"* anchor (R2).** We conduct experiments where the model was trained on a dataset with synthetic complete modalities to mimic the complete-modality scenario. Figure 5 shows the comparison of the variant without calibration and the one with complete modalities on the MSR-VTT. Compared to the original performance, training on complete modalities can best maintain the alignment, achieving results around or above the red line. CalMRL outperforms the missing-modality variant in most cases and even the "ideal" (complete-modality training) in the VATEX dataset. We attribute it to the mitigation of the inherent noise in the full datasets. In contrast, imposing learning on

un-match examples leads to certain negative effects.

**Training stability (R3).** We plot the curves of the training loss $\mathcal{L}_{\text{rep}}$ for different models in Figure 7. VAST, GRAM, and TRIANGLE all utilize text-anchored contrastive learning, which leads to relatively larger losses, while CalMRL follows PMRL, optimizing with smaller ones. Overall, we can observe that the loss curve of CalMRL is more stable and exhibits a steadier decreasing trend compared to the others. Despite employing bi-step learning for missing modalities, CalMRL is capable of managing the training and achieving convergence.

**Case visualization (R3).** To intuitively understand the inner workings of CalMRL, we visualize the representations as points within a 2D plane to analyze their distributions. As shown in Figure 6, VAST, GRAM, and TRIANGLE demonstrate distinct, well-separated regions for each modality. While PMRL successfully aligns the vision and text representations, it excludes the audio representation with a clear dividing line. In contrast, CalMRL alleviates this separation, pulling the multiple modalities closer together. We attribute this improvement to the alignment calibration, the core design philosophy of CalMRL, which effectively prevents the segregation of different modal representations.

## 5. Conclusion

In this paper, we presented CalMRL, a calibrated multimodal representation learning framework for handling missing modalities. We theoretically revealed the anchor shift phenomenon and proposed a generative imputation mechanism to reconstruct representations and calibrate alignment. Through a bi-step optimization with closed-form inference and iterative refinement, CalMRL achieves stable convergence and strong compatibility with existing alignment paradigms. Experiments across diverse benchmarks verify its superiority. Looking ahead, future explorations can extend our method to broader applications and focus on: calibrating the shift without imputation, incorporating more datasets to enhance multimodal models, and interpreting multimodal connections to inspire further advancement.

## Impact Statement

This paper presents work whose goal is to advance the field of multimodal learning by addressing the "anchor shift" problem in modality missing scenarios. By providing a theoretical and practical framework for handling missing modalities, our work facilitates the development of more flexible AI systems that can operate in data-constrained environments. While the use of generative models to impute missing data carries the general risk of perpetuating biases inherent in the training distribution, we believe there are no specific societal or ethical consequences of our work that must be uniquely highlighted here beyond the standard considerations for the field.

## Acknowledgement

This research/project is supported by the National Research Foundation, Singapore under its National Large Language Models Funding Initiative (AISG Award No: AISG-NMLP-2024-002). Any opinions, findings and conclusions or recommendations expressed in this material are those of the author(s) and do not reflect the views of National Research Foundation, Singapore.

Jiaheng Wei is partially supported by Guangdong Provincial Key Lab of Integrated Communication, Sensing and Computation for Ubiquitous Internet of Things (No. 2023B1212010007). Shuo Yang is supported by the Shenzhen Fundamental Research Program (JCYJ20250604145514018), Guangdong Basic and Applied Basic Research Foundation (General Program, No. 2026A1515011557), and the NSFC Young Scientists Fund (No. 62506096). Xiu Su is supported by the National Natural Science Foundation of China (No. 62406347).

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

# Appendix

## Contents

# A. Theoretical Analysis

In this section, we detail the proof of Theorem 1 (*cf.*, Appendix A.1), the derivation of the closed-form solution of $\widehat{\theta}$ (*cf.*, Appendix A.2), the anchor shift after calibration (*cf.*, Appendix A.3), and the convergence analysis (*cf.*, Appendix A.4).

## A.1. Proof of Theorem 1

*Recall.* Let $\mathbf{u}_1$ and $\mathbf{u}_1^{\Omega}$ be the leading left singular vectors of the full multimodal matrix $\mathbf{Z}$ and its observed submatrix $\mathbf{Z}^{\Omega}$, respectively. Define $\sigma_1 = \|\mathbf{Z}\|_2$, $\sigma_1^{\Omega} = \|\mathbf{Z}^{\Omega}\|_2$, and $\eta := \sqrt{\sum_{m \in \bar{\Omega}} \langle \mathbf{u}_1^{\Omega}, \mathbf{z}^m \rangle^2}$. Then the anchor shift satisfies

$$\sqrt{2\left(1 - \frac{\sigma_1^{\Omega} + \eta^2}{\sigma_1}\right)} \;\leq\; \underbrace{\|\mathbf{u}_1 - \mathbf{u}_1^{\Omega}\|}_{\|\Delta\|} \;\leq\; \frac{\sqrt{2}\,\|\mathbf{Z}^{\bar{\Omega}}\|_2}{\sqrt{\lambda_1} - \sqrt{\lambda_2}}.$$

*Proof.* The multimodal representation is expressed by the concatenation of different unimodal representations:

$$\mathbf{Z} = [\mathbf{z}^{m_1}, \mathbf{z}^{m_2}, \dots, \mathbf{z}^{m_k}] \in \mathbb{R}^{d \times k}, \tag{11}$$

$$= \mathbf{U}\boldsymbol{\Sigma}\mathbf{V}^{\top}, \tag{12}$$

$$\mathbf{U} = [\mathbf{u}_1, \dots, \mathbf{u}_d] \in \mathbb{R}^{d \times d}, \tag{13}$$

$$\boldsymbol{\Sigma} = \mathrm{diag}(\sigma_1, \dots, \sigma_{\min\{d,k\}}), \tag{14}$$

where $\sigma_1 \geq \sigma_2 \geq \cdots \geq 0$. We define the virtual anchor as $\mathbf{u}_1$, which is the consensus direction in the spanned by multimodalities.

According to the Davis–Kahan Theorem, we have:

$$\sin \angle(\mathbf{u}_1, \mathbf{u}_1^{\Omega}) \leq \frac{\|\mathbf{Z}^{\bar{\Omega}}\|_2}{\delta}, \tag{15}$$

where $\delta = \sigma_1 - \sigma_2$ represents the spectral gap and $\|\mathbf{Z}^{\bar{\Omega}}\|_2 = \sigma_{\max}(\mathbf{Z}^{\bar{\Omega}})$ denotes the spectral norm of the submatrix associated with the missing modalities.

$$\|\boldsymbol{\Delta}\| := \|\mathbf{u}_1 - \mathbf{u}_1^{\Omega}\| = [(\mathbf{u}_1 - \mathbf{u}_1^{\Omega})^{\top}(\mathbf{u}_1 - \mathbf{u}_1^{\Omega})]^{-\frac{1}{2}} \tag{16}$$

$$= [\|\mathbf{u}_1\|^2 + \|\mathbf{u}_1^{\Omega}\|^2 - 2\mathbf{u}_1^{\top}\mathbf{u}_1^{\Omega}]^{-\frac{1}{2}} \tag{17}$$

$$= [2 - 2\cos\theta]^{-\frac{1}{2}} \tag{18}$$

$$= [4\sin^2(\theta/2)]^{-\frac{1}{2}} \qquad (1 - \cos\theta = 2\sin^2(\theta/2)) \tag{19}$$

$$= 2|\sin(\theta/2)| = 2\sin(\theta/2) \qquad (\theta \leq \pi/2) \tag{20}$$

$$\leq \sqrt{2}\sin(\theta) \tag{21}$$

$$\leq \frac{\sqrt{2}\|\mathbf{Z}^{\bar{\Omega}}\|_2}{\sigma_1 - \sigma_2} \qquad (\text{Eq. (15)}) \tag{22}$$

$$= \frac{\sqrt{2}\|\mathbf{Z}^{\bar{\Omega}}\|_2}{\sqrt{\lambda_1} - \sqrt{\lambda_2}} \qquad (\sigma_1(\mathbf{Z}) = \lambda_1(\mathbf{G})) \tag{23}$$

$$= \frac{\sqrt{2}\|\mathbf{Z}^{\bar{\Omega}}\|_2}{\mathcal{A}} \qquad (\mathcal{A} := \sqrt{\lambda_1} - \sqrt{\lambda_2}) \tag{24}$$

$$\leq \frac{\sqrt{2|\bar{\Omega}|}}{\mathcal{A}} \qquad (\|\mathbf{Z}^{\bar{\Omega}}\|_2 \leq \sqrt{|\bar{\Omega}|}) \tag{25}$$

This upper bound indicates that a better alignment can lead to better robustness for the case of missing modality.

Now we derive the lower bound as follows:

$$\|\boldsymbol{\Delta}\| := \|\mathbf{u}_1 - \mathbf{u}_1^\Omega\| = [2 - 2\cos\theta]^{1/2} \tag{26}$$

$$\geq [2(1 - \cos^2\theta)]^{1/2} \tag{27}$$

$$= [2(1 - |\langle \mathbf{u}_1, \mathbf{u}_1^\Omega \rangle|^2)]^{1/2} \tag{28}$$

$$\geq [2 - 2\frac{\|\mathbf{u}_1^{\Omega\top}\mathbf{Z}\|_2^2}{\sigma_1}]^{1/2} \tag{29}$$

$$= [2 - 2\frac{\|\mathbf{u}_1^{\Omega\top}\mathbf{Z}^\Omega\|_2^2 + \|\mathbf{u}_1^{\Omega\top}\mathbf{Z}^{\bar\Omega}\|_2^2}{\sigma_1}]^{1/2} \tag{30}$$

$$= [2 - 2\frac{(\sigma_1^\Omega)^2 + \eta^2}{\sigma_1}]^{1/2} \qquad (\eta := \sqrt{\sum_{m\in\bar\Omega}\langle \mathbf{u}_1^\Omega, \mathbf{z}^m \rangle^2}) \tag{31}$$

This lower bound reveals that the anchor shift cannot be arbitrarily small. It is fundamentally limited by how much the missing modalities $\bar\Omega$ contribute to the leading singular direction. $\qquad\square$

## A.2. Resolving $\widehat{\theta}$

We consider the joint distribution of $\mathbf{z}$ and $\boldsymbol{\beta}$:

$$\begin{bmatrix}\boldsymbol{\beta}_n \\ \mathbf{z}_n\end{bmatrix} \sim \mathcal{N}\left(\begin{bmatrix}\mathbf{0} \\ \boldsymbol{\mu}\end{bmatrix}, \begin{bmatrix}\mathbf{I} & \mathbf{W}^\top \\ \mathbf{W} & \mathbf{W}\mathbf{W}^\top + \boldsymbol{\Sigma}\end{bmatrix}\right), \tag{32}$$

$$\boldsymbol{\mu} = [\boldsymbol{\mu}^{1\top}, \ldots, \boldsymbol{\mu}^{|\Omega|\top}]^\top \tag{33}$$

$$\mathbf{W} = [\mathbf{W}^{1\top}, \ldots, \mathbf{W}^{|\otimes|\top}]^\top \tag{34}$$

$$\boldsymbol{\Sigma} = \text{diag}((\sigma^1)^2\mathbf{I}, \ldots, (\sigma^{|\Omega|})^2\mathbf{I}) \tag{35}$$

The condition distribution is $p(\mathbf{z} \mid \mathbf{x}) = \mathcal{N}(\mathbf{z}; \mathbf{m}, \mathbf{V})$.

$$\mathbf{V} = \boldsymbol{\Sigma}_{\boldsymbol{\beta}} - \boldsymbol{\Sigma}_{\boldsymbol{\beta}\mathbf{z}}\boldsymbol{\Sigma}_{\mathbf{z}}^{-1}\boldsymbol{\Sigma}_{\mathbf{z}\boldsymbol{\beta}} \tag{36}$$

$$= \mathbf{I} - \mathbf{W}^\top(\mathbf{W}\mathbf{W}^\top + \boldsymbol{\Sigma})^{-1}\mathbf{W} \tag{37}$$

$$= \mathbf{I} - \mathbf{W}^\top[\boldsymbol{\Sigma}^{-1} - \boldsymbol{\Sigma}^{-1}\mathbf{W}(\mathbf{I}_K + \mathbf{W}^\top\boldsymbol{\Sigma}^{-1}\mathbf{W})^{-1}\mathbf{W}^\top\boldsymbol{\Sigma}^{-1}]\mathbf{W} \qquad \text{(Woodbury matrix identity (Woodbury, 1949))}$$

$$= \mathbf{I} - \mathbf{A} + \mathbf{A}(\mathbf{I}_K + \mathbf{A})^{-1}\mathbf{A} \quad (\mathbf{A} = \mathbf{W}^\top\boldsymbol{\Sigma}^{-1}\mathbf{W}) \tag{38}$$

$$= (\mathbf{I} + \mathbf{W}^\top\boldsymbol{\Sigma}^{-1}\mathbf{W})^{-1} \quad (\mathbf{A}(\mathbf{I}+\mathbf{A})^{-1} = \mathbf{I} - (\mathbf{I}+\mathbf{A})^{-1}) \tag{39}$$

$$= \left[\mathbf{I} + \sum_{m\in\Omega}\frac{1}{(\sigma^m)^2}\mathbf{W}^{m\top}\mathbf{W}^m\right]^{-1} \tag{40}$$

$$\mathbf{m} = \mathbb{E}[\mathbf{z}_n \mid \mathbf{x}_n] \tag{41}$$

$$= \mathbf{0} + \mathbf{W}^\top(\mathbf{W}\mathbf{W}^\top + \boldsymbol{\Sigma})^{-1}(\mathbf{z} - \boldsymbol{\mu}) \tag{42}$$

$$= (\mathbf{I} + \mathbf{W}^\top\boldsymbol{\Sigma}^{-1}\mathbf{W})^{-1}\mathbf{W}^\top\boldsymbol{\Sigma}^{-1}(\mathbf{z} - \boldsymbol{\mu}) \tag{43}$$

$$= \mathbf{V}\mathbf{W}^\top\boldsymbol{\Sigma}^{-1}(\mathbf{z} - \boldsymbol{\mu}) \tag{44}$$

$$= \mathbf{V}\sum_{m\in\Omega}\frac{1}{(\sigma^m)^2}\mathbf{W}^{m\top}(\mathbf{z}^m - \boldsymbol{\mu}^m) \tag{45}$$

Therefore, we have:

$$\begin{cases}\mathbf{V} = \left[\mathbf{I} + \sum_{m\in\Omega}\frac{1}{(\sigma^m)^2}\mathbf{W}^{m\top}\mathbf{W}^m\right]^{-1}, \\ \mathbf{m} = \mathbf{V}\sum_{m\in\Omega}\frac{1}{(\sigma^m)^2}\mathbf{W}^{m\top}(\mathbf{z}^m - \boldsymbol{\mu}^m),\end{cases} \tag{46}$$

The objective is updated as:

$$\mathbb{E}[\log p(\mathbf{z}, \boldsymbol{\beta}|\widehat{\boldsymbol{\theta}})] = \mathbb{E}\Big[\log p(\boldsymbol{\beta}) + \sum_{m \in \Omega} \log p(\mathbf{z}^m \mid \boldsymbol{\beta}, \widehat{\boldsymbol{\theta}})\Big] \tag{47}$$

$$= \mathbb{E}\Big[-\frac{1}{2}\boldsymbol{\beta}^\top \boldsymbol{\beta} + \sum_{m \in \Omega}\Big[-\frac{d}{2}\log(\sigma^m)^2 - \frac{1}{2(\sigma^m)^2}\mathbb{E}[\|\mathbf{z}^m - \boldsymbol{\mu}^m - \mathbf{W}^m\boldsymbol{\beta}\|^2]\Big] + c\Big] \tag{48}$$

Let us compute the closed-form solution of $\boldsymbol{\mu}^m$ and $\mathbf{W}^m$ according to its related terms.

$$\mathbb{E}\|\mathbf{z}^m - \boldsymbol{\mu}^m - \mathbf{W}^m\boldsymbol{\beta}\|^2 = \|\mathbf{z}^m - \boldsymbol{\mu}^m - \mathbf{W}^m\mathbf{m}\|^2 + \mathrm{Tr}[\mathbf{W}^{m\top}\mathbf{W}^m\mathbf{V}]$$

$$\rightarrow \boldsymbol{\mu}^m = \frac{1}{N}\sum_n(\mathbf{z}_n^m - \mathbf{W}^m\mathbf{m}_n) \tag{49}$$

$$\rightarrow \mathbf{W}^m = \Big(\sum_{n=1}^N(\mathbf{z}_n^m - \boldsymbol{\mu}^m)\mathbf{m}_n^\top\Big)\Big(\sum_{n=1}^N \mathbb{E}[\boldsymbol{\beta}_n\boldsymbol{\beta}_n^\top]\Big)^{-1} \tag{50}$$

We can also calculate the solution for $\sigma_m^2$:

$$\sigma_m^2 = \frac{1}{Nd}\sum_{n=1}^N\big[\|\mathbf{z}_n^m - \boldsymbol{\mu}^m - \mathbf{W}^m\mathbf{m}_n\|^2 + \mathrm{Tr}[\mathbf{W}^{m\top}\mathbf{W}^m\mathbf{V}]\big] \tag{51}$$

Therefore, for missing modalities $m' \in \mathcal{M}/\Omega$ conditioned on the previous observations, we have:

$$\mathbf{z}^{m'} = \mathbf{W}^{m'}\bar{\boldsymbol{\beta}} + \boldsymbol{\mu}^{m'} + \bar{\boldsymbol{\epsilon}}^m \tag{52}$$

$$= \mathbf{W}^{m'}\mathbb{E}[\boldsymbol{\beta}|X] + \boldsymbol{\mu}^{m'} + \mathbb{E}[\boldsymbol{\epsilon}^m|X] \tag{53}$$

$$= \mathbf{W}^{m'}\mathbf{m} + \boldsymbol{\mu}^{m'} \tag{54}$$

## A.3. Alleviating Anchor Shift

Let $\widehat{\mathbf{Z}}^{\bar{\Omega}} = [\mathbf{W}^{m'}\mathbf{m} + \mu^{m'}|m \in \mathcal{M}/\Omega]$.

$$\|\Delta_{\text{cal}}\| = \|\mathbf{u}_1 - \mathbf{u}_1^{\text{cal}}\| \tag{55}$$

$$\leq \frac{\sqrt{2}\|[\mathbf{Z}^\Omega, \widehat{\mathbf{Z}}^{\bar{\Omega}}] - \mathbf{Z}\|_2}{\sigma_1 - \sigma_2} \tag{56}$$

$$\leq \frac{\sqrt{2}\|[\mathbf{Z}^\Omega, \widehat{\mathbf{Z}}^{\bar{\Omega}}] - \mathbf{Z}\|_F}{\sigma_1 - \sigma_2} \tag{57}$$

$$\leq \frac{\sqrt{2|\bar{\Omega}|}\varepsilon}{\sigma_1 - \sigma_2} \qquad (\|\hat{\mathbf{z}}^{m'} - \hat{\mathbf{z}}^{m'}\|_2 \leq \varepsilon, \forall m' \in \bar{\Omega})$$

Here $\varepsilon$ represents the imputation error at the representation level. From Theorem 1, we have:

$$\|\Delta\| \geq \sqrt{2\big(1 - \frac{\sigma_1^\Omega + \eta^2}{\sigma_1}\big)}. \tag{58}$$

Therefore, we can derive the condition for $\|\Delta_{\text{cal}}\| \leq \|\Delta\|$ as follows:

$$\varepsilon < \frac{\sigma_1 - \sigma_2}{\sqrt{|\bar{\Omega}|}} \cdot \sqrt{1 - \frac{\sigma_1^\Omega + \eta^2}{\sigma_1}}. \tag{59}$$

This also provides a consistent conclusion to alleviate the anchor shift. The calibration is beneficial, especially in the case of strong alignment among modalities ($\uparrow \sigma_1 - \sigma_2$) and greater original shift ($\downarrow \frac{\sigma_1^\Omega + \eta^2}{\sigma_1}$).

## A.4. Convergence Analysis.

*Recall. Let $\widehat{\boldsymbol{\theta}}^{(t)}$ be the parameters at iteration $t$ applied to the observed-data log-likelihood $L(\widehat{\boldsymbol{\theta}}) = \sum_n \log p(\mathbf{z}_n^\Omega \mid \widehat{\boldsymbol{\theta}})$ under the generative model in Eq. (2). Given the solution of $q$ and $\widehat{\boldsymbol{\theta}}$, $L(\widehat{\boldsymbol{\theta}}^{(t+1)}) \geq L(\widehat{\boldsymbol{\theta}}^{(t)})$.*

*Proof.* Let $L(\widehat{\boldsymbol{\theta}}^{(t+1)})$ be the log-likelihood for parameters at $t+1$ step.

$$L(\widehat{\boldsymbol{\theta}}^{(t+1)}) = \sum_{n=1}^{N} \log p(\mathbf{z}_n^\Omega | \widehat{\boldsymbol{\theta}}^{(t+1)}) \tag{60}$$

$$= \sum_{n=1}^{N} \log \int p(\mathbf{z}_n^\Omega, \boldsymbol{\beta}_n | \widehat{\boldsymbol{\theta}}^{(t+1)}) d\boldsymbol{\beta}_n \tag{61}$$

$$= \sum_{n=1}^{N} \log \mathbb{E}_{q_n} \left[ \frac{p(\mathbf{z}_n^\Omega, \boldsymbol{\beta}_n | \widehat{\boldsymbol{\theta}}^{(t+1)})}{q_n^{(t+1)}(\boldsymbol{\beta}_n)} \right] \qquad (\mathbf{m}_n, \mathbf{V}_n \leftarrow \widehat{\boldsymbol{\theta}}^{(t)})$$

$$\geq \sum_{n=1}^{N} \mathbb{E}_{q_n} \left[ \log \frac{p(\mathbf{z}_n^\Omega, \boldsymbol{\beta}_n | \widehat{\boldsymbol{\theta}}^{(t+1)})}{q_n^{(t+1)}(\boldsymbol{\beta}_n)} \right] \tag{62}$$

$$\geq \sum_{n=1}^{N} \mathbb{E}_{q_n} \left[ \log \frac{p(\mathbf{z}_n^\Omega, \boldsymbol{\beta}_n | \widehat{\boldsymbol{\theta}}^{(t)})}{q_n^{(t+1)}(\boldsymbol{\beta}_n)} \right] := \mathcal{Q} \qquad (\widehat{\boldsymbol{\theta}}^{(t+1)} = \arg\max_{\widehat{\boldsymbol{\theta}}} \mathcal{Q}(q^{(t+1)}, \widehat{\boldsymbol{\theta}}))$$

$$= L(\widehat{\boldsymbol{\theta}}^{(t)}) - \underbrace{\mathrm{KL}[q_n^{(t+1)}(\boldsymbol{\beta}_n) \| p(\boldsymbol{\beta}_n | \mathbf{z}_n^\Omega, \widehat{\boldsymbol{\theta}})]}_{0} \qquad (q(\boldsymbol{\beta}) = p(\boldsymbol{\beta}|\mathbf{z}, \widehat{\boldsymbol{\theta}}))$$

$$= L(\widehat{\boldsymbol{\theta}}^{(t)}) \tag{63}$$

Therefore, we have $L(\widehat{\boldsymbol{\theta}}^*) \geq \cdots \geq L(\widehat{\boldsymbol{\theta}}^{(t+1)}) \geq L(\widehat{\boldsymbol{\theta}}^{(t)}) \geq \cdots \geq L(\widehat{\boldsymbol{\theta}}^{(0)})$, indicating our method can converges to a stationary point of $L(\widehat{\boldsymbol{\theta}}^*)$. □

## B. Related Work

**Multimodal representation learning.** To integrate multiple modalities, researchers align them within a unified latent space, where one unimodal representation can be retrieved from another when they correspond to a common instance (Lu, 2023; Zong et al., 2024; Xu et al., 2023; Liu et al., 2025b; Jamal & Mohareri, 2025; Li et al., 2025b; Shen et al., 2025; Chen et al., 2026; Luo et al., 2025; Liu et al., 2025a). For example, CLIP (Radford et al., 2021) pioneers grounding vision in language space through pairwise contrastive learning (Chen et al., 2020b; Zolfaghari et al., 2021; Zhang et al., 2021; Shao et al., 2024), inspiring a series of methods for connecting bimodalities (Guzhov et al., 2022; Wu et al., 2022; Elizalde et al., 2023; Zhang et al., 2022b; Xu et al., 2021; Luo et al., 2022). Recent work goes beyond pairs, adapting the CLIP paradigm to incorporate more modalities (Guo et al., 2023; Wang et al., 2024b;c; Lyu et al., 2024), exemplified by ImageBind (Girdhar et al., 2023) and LanguageBind (Zhu et al., 2023). Large-scale data collection (Bain et al., 2021; Nagrani et al., 2022; Zhao et al., 2024; Wang et al., 2024a; Ju et al., 2024; Fu et al., 2024; Nan et al., 2025) and architectural advancements (Li et al., 2022; Chen et al., 2023a;b; Srivastava & Sharma, 2024; Lei et al., 2024b; Han et al., 2024), like VAST (Chen et al., 2023b), have furthered progress, though the learning objective remains constrained to pairwise alignment, *i.e.*, aligning one modality with a predefined anchor. GRAM (Cicchetti et al., 2025b) proposes to learn multimodal representations simultaneously in a geometric manner by minimizing the volume of a parallelotope spanned by the modality vectors. TRIANGLE (Cicchetti et al., 2025a) builds on this by minimizing the designed area for three modalities. PMRL (Liu et al., 2026) establishes principled foundations and proposes maximizing the largest eigenvalue for anchor-free alignment. However, to support its optimization, all modalities must be present to ensure unbiased estimation, unlike prevailing datasets that typically include only two modalities and thus have missing modalities. To endow this emerging paradigm with greater flexibility, we propose calibrating the incomplete alignment with missing modalities for multimodal representation learning.

**Learning with missing modalities.** This setting indicates the incomplete data with respect to modalities, necessitating the model to perform nearly as well as when all modalities are present (Wu et al., 2024; Ma et al., 2022). The prevailing

*Table 4.* The statistics of datasets.

| Benchmark | Modalities | | | #Train | #Test |
| | Video | Audio | Text | | |
|---|---|---|---|---|---|
| MSR-VTT | ✓ | - | ✓ | 180,000 | 1,000 |
| DiDeMo | ✓ | - | ✓ | - | 1,003 |
| ActivityNet | ✓ | - | ✓ | - | 3,987 |
| VATEX | ✓ | - | ✓ | - | 1,225 |
| AudioCaps | - | ✓ | ✓ | 45,178 | 704 |
| Clotho | - | ✓ | ✓ | - | 1,045 |

way is modality imputation, which involves filling in the missing information by composing existing modalities and generating absent modalities (Cai et al., 2018). Typically, this focuses on generating high-quality raw data for the missing modalities (Cai et al., 2018; Wang et al., 2023c) (*e.g.*, using an auxiliary adversarial loss to generate missing images (Cai et al., 2018)) or, more commonly, crafting the representations in the latent space based on the observed ones (Zhao et al., 2021; Ma et al., 2021; Zhang et al., 2022a; Wang et al., 2023a; Jin et al., 2023; Tang et al., 2024; Chen et al., 2024; Yun et al., 2024) (*e.g.*, indexing from previous multimodal interactions (Zhang et al., 2022a; Yun et al., 2024)). Some studies also employ advanced models for distillation to yield superior modality representations (Ke et al., 2025; Lao et al., 2025; Wang et al., 2023b). Another research line is to design a specialized fusion module to take advantage of available modalities (Xu et al., 2024; Li et al., 2025a; Reza et al., 2024). For instance, SimMLM (Li et al., 2025a) introduces a gating network to weigh the contribution of experts corresponding to each modality, and Reza et al. (2024) designs a modulation function to compensate for the missing modalities. However, the above methods are mainly specialized for downstream tasks. Learning better multimodal representations with incomplete modalities is not addressed in the fundamental perspective of multimodal alignment. This motivates this work, especially for the emergence of multimodalities in diverse applications.

We compare CalMRL against two research lines (*i.e.*, classic pCCA and multimodal VAEs) from a technical perspective to highlight their parallels. *Classical methods:* While CalMRL's generative model shares a linear-Gaussian form, we apply it as a lightweight calibration module in the encoded representation space, coupled with a modern SVD-based alignment objective. Classical pCCA/GFA operate on raw features (Bach & Jordan, 2005; Klami et al., 2014), whereas CalMRL delegates nonlinearity to pretrained encoders and uses the linear model solely for cross-modal imputation. This decomposition enables our anchor-shift analysis (Theorem 1, Corollary 3), which is specific to the alignment-via-Gramian paradigm used in PMRL (Liu et al., 2026) and GRAM (Cicchetti et al., 2025b). *Multimodal VAEs:* MVAE (Wu & Goodman, 2018) and MoPoE (Sutter et al., 2021) use amortized variational inference with nonlinear encoder-decoder pairs end-to-end. CalMRL instead uses exact, closed-form inference, trading decoder expressivity for analytical guarantees and training simplicity. Notably, the PoE structure in our posterior (Eq. 5) is structurally related to MVAE's Gaussian product-of-experts, but utilizes global (rather than instance-dependent) precision contributions.

## C. Implementation Details

### C.1. Datasets

We employ the training dataset *VAST-150K* (Cicchetti et al., 2025b) at the warm-up stage to avoid a cold start. This dataset is a downsized version of VAST-27M (Chen et al., 2023b) and used for previous works (Cicchetti et al., 2025b; Liu et al., 2026). VAST-27M involves four modalities, *i.e.*, video, audio, caption, and subtitle for each example. We also adopt the training splits of *MSR-VTT* (Chen & Dolan, 2011) (around 180K, where 20 captions for each video) for vision-text and *AudioCaps* (Kim et al., 2019) (around 45K) for audio-text for the missing modality training. For benchmarking, we adopt several datasets with their test splits, including vision-text datasets *MSR-VTT* (Chen & Dolan, 2011), *DiDeMo* (Anne Hendricks et al., 2017), *ActivityNet* (Krishna et al., 2017), and *VATEX* (Wang et al., 2019), and audio-text datasets *AudioCaps* (Kim et al., 2019) and *Clotho* (Drossos et al., 2020). All videos are sampled into 8 frames. For the testing splits, the complete modalities are provided (Chen et al., 2023b). The statistics of these benchmarks are shown in Table 4.

*Table 5.* Multimodal retrieval results (%) for models trained on full datasets and sole datasets with respect to Recall@5.

| | MSR-VTT | | DiDeMo | | ActivityNet | | VATEX | | AudioCaps | | Clotho | |
|---|---|---|---|---|---|---|---|---|---|---|---|---|
| | T→V | V→T | T→V | V→T | T→V | V→T | T→V | V→T | T→A | A→T | T→A | A→T |
| VAST ↑ | 76.8 | 81.8 | 74.6 | 75.6 | 80.2 | 80.8 | 95.7 | 95.8 | 81.5 | 84.7 | 49.0 | 45.1 |
| GRAM ↑ | 78.7 | 81.1 | 75.3 | 75.7 | 80.0 | 80.3 | 96.7 | 96.8 | 82.4 | 83.9 | 47.2 | 43.6 |
| Triangle ↑ | 78.4 | 79.9 | 73.9 | 74.9 | 79.3 | 79.4 | 94.5 | 95.3 | 80.4 | 84.1 | 42.8 | 45.0 |
| PMRL ↑ | 79.6 | 78.8 | 74.9 | 76.5 | 80.9 | 81.3 | 95.6 | 96.2 | 81.7 | 83.5 | 49.0 | 47.6 |
| CalMRL ↑ | 83.0 | 80.5 | 75.6 | 76.2 | 80.7 | 81.6 | 96.7 | 96.7 | 80.8 | 82.4 | 47.0 | 46.0 |
| VAST↑$^{AT}$ | 71.8 | 72.1 | 65.8 | 68.4 | 72.6 | 74.2 | 94.4 | 94.1 | 81.2 | 82.5 | 47.0 | 46.5 |
| GRAM↑$^{AT}$ | 67.1 | 73.0 | 70.3 | 70.2 | 73.2 | 74.0 | 66.7 | 92.8 | 84.7 | 85.1 | 49.0 | 46.6 |
| Triangle↑$^{AT}$ | 76.1 | 74.3 | 71.6 | 73.6 | 76.2 | 76.4 | 95.2 | 95.2 | 79.3 | 81.4 | 41.1 | 43.5 |
| PMRL↑$^{AT}$ | 77.1 | 73.8 | 70.4 | 72.3 | 77.0 | 76.1 | 94.9 | 94.8 | 84.5 | 85.2 | 49.9 | 48.0 |
| CalMRL↑$^{AT}$ | 76.9 | 74.2 | 71.7 | 72.8 | 78.1 | 76.7 | 95.3 | 94.5 | 82.8 | 83.1 | 45.8 | 46.5 |
| VAST↑$^{VT}$ | 77.0 | 82.1 | 76.8 | 76.8 | 82.0 | 81.8 | 96.0 | 96.7 | 64.3 | 66.2 | 32.8 | 34.0 |
| GRAM↑$^{VT}$ | 79.9 | 82.4 | 76.6 | 78.1 | 80.5 | 80.6 | 97.1 | 96.8 | 66.2 | 67.9 | 34.5 | 34.4 |
| Triangle↑$^{VT}$ | 78.8 | 81.2 | 75.6 | 75.8 | 81.0 | 80.0 | 95.3 | 95.6 | 61.5 | 64.2 | 35.2 | 34.2 |
| PMRL↑$^{VT}$ | 79.8 | 81.3 | 76.8 | 76.6 | 82.3 | 81.3 | 96.3 | 96.2 | 63.1 | 66.9 | 38.7 | 35.6 |
| CalMRL ↑$^{VT}$ | 82.5 | 80.9 | 76.9 | 77.9 | 82.1 | 81.0 | 96.5 | 97.3 | 64.9 | 66.6 | 39.0 | 36.1 |

*Table 6.* Multimodal retrieval results (%) for models trained on full datasets and sole datasets with respect to Recall@10.

| | MSR-VTT | | DiDeMo | | ActivityNet | | VATEX | | AudioCaps | | Clotho | |
|---|---|---|---|---|---|---|---|---|---|---|---|---|
| | T→V | V→T | T→V | V→T | T→V | V→T | T→V | V→T | T→A | A→T | T→A | A→T |
| VAST ↑ | 81.4 | 85.7 | 79.2 | 80.2 | 86.8 | 88.5 | 97.3 | 97.1 | 89.9 | 92.9 | 60.4 | 59.5 |
| GRAM ↑ | 86.0 | 86.9 | 80.2 | 81.0 | 86.9 | 87.9 | 98.2 | 98.4 | 90.8 | 94.2 | 60.4 | 57.9 |
| Triangle ↑ | 83.8 | 85.1 | 79.1 | 79.6 | 86.8 | 87.4 | 95.8 | 97.1 | 90.1 | 93.6 | 57.0 | 59.1 |
| PMRL ↑ | 84.5 | 85.2 | 79.6 | 81.6 | 88.0 | 89.0 | 97.0 | 97.3 | 90.3 | 92.8 | 58.9 | 59.7 |
| CalMRL ↑ | 87.3 | 87.1 | 80.5 | 82.0 | 87.9 | 89.5 | 98.1 | 98.0 | 88.2 | 91.3 | 58.5 | 59.1 |
| VAST↑$^{AT}$ | 78.4 | 79.8 | 72.3 | 76.5 | 79.8 | 82.5 | 95.8 | 95.4 | 91.2 | 92.9 | 60.8 | 60.0 |
| GRAM↑$^{AT}$ | 72.4 | 79.1 | 74.8 | 77.2 | 80.0 | 82.5 | 67.1 | 94.9 | 93.3 | 94.3 | 61.5 | 60.6 |
| Triangle↑$^{AT}$ | 82.6 | 81.1 | 77.3 | 79.3 | 83.1 | 84.7 | 96.6 | 96.3 | 88.8 | 90.2 | 55.1 | 58.1 |
| PMRL↑$^{AT}$ | 82.7 | 80.7 | 75.9 | 78.9 | 84.1 | 85.2 | 96.1 | 96.3 | 92.0 | 92.8 | 61.1 | 61.8 |
| CalMRL↑$^{AT}$ | 83.3 | 81.1 | 77.2 | 78.7 | 85.3 | 85.7 | 96.2 | 96.6 | 90.5 | 91.5 | 59.0 | 59.0 |
| VAST↑$^{VT}$ | 82.9 | 86.2 | 80.9 | 82.7 | 88.6 | 89.2 | 97.4 | 97.6 | 75.9 | 79.3 | 43.9 | 44.6 |
| GRAM↑$^{VT}$ | 86.9 | 88.0 | 80.9 | 82.9 | 87.5 | 88.5 | 98.4 | 98.0 | 76.6 | 81.4 | 45.5 | 43.3 |
| Triangle↑$^{VT}$ | 84.3 | 86.8 | 80.2 | 80.5 | 87.9 | 88.1 | 97.5 | 97.5 | 73.6 | 78.7 | 47.0 | 45.5 |
| PMRL↑$^{VT}$ | 85.6 | 87.1 | 81.4 | 82.5 | 88.9 | 88.8 | 97.8 | 97.4 | 77.1 | 79.8 | 49.4 | 46.8 |
| CalMRL ↑$^{VT}$ | 86.3 | 87.6 | 81.9 | 83.2 | 89.3 | 88.9 | 98.4 | 98.6 | 76.6 | 79.1 | 49.1 | 45.8 |

## C.2. Model Architecture

We build our model based on VAST (Chen et al., 2023b) to ensure a fair comparison rather than advancing the architecture design. Specifically, we construct the vision encoder with EVAClip-ViT-G (Sun et al., 2023), where the vision resolution is set to 224×224 pixels. BERT is utilized to implement the text encoder with a maximum caption length limited to 40. For subtitles, the maximum length is extended to 70. BEATs model (Chen et al., 2023c) is adopted for audio encoding. Each audio is preprocessed into 63 mel-frequency bins, outputting 1024 frames. We also built different multimodal alignment methods with ImageBind (Girdhar et al., 2023) as the backbone. To facilitate efficient fine-tuning, we appended an additional projector after the backbone. Wherein, we implement the VAST baseline using multiple contrastive losses to align the various modalities. To ensure a fair comparison and rigorous validation, we maintain the identical model architectures and hyperparameter settings across VAST, GRAM, TRIANGLE, PMRL, and CalMRL.

*Table 7.* Multimodal retrieval results (%) for different generative methods adapted for the missing modality question.

| | MSRVTT | ActivityNet | DiDeMo | VATEX | AudioCaps | Clotho | **Avg.** |
|---|---|---|---|---|---|---|---|
| **CalMRL (Ours)** | 61.1 | 55.4 | 57.1 | 81.3 | 50.1 | 23.8 | **54.8** |
| MVAE (Wu & Goodman, 2018) | 61.0 | 55.0 | 55.9 | 80.8 | 48.4 | 23.0 | 54.0 |
| MoPoE (Sutter et al., 2021) | 61.0 | 55.2 | 55.6 | 80.7 | 48.7 | 23.5 | 54.1 |
| SMIL (Ma et al., 2021) | 60.2 | 54.8 | 55.0 | 80.6 | 49.4 | 24.2 | 54.0 |
| Knowledge Bridger (Ke et al., 2025) | 60.9 | 54.7 | 55.1 | 80.8 | 48.9 | 24.5 | 54.2 |

*Table 8.* Performance comparison on different hyperparameters, including $\tau$, $\tau'$ and $\alpha$

| $\tau$ | 0.05 | 0.1 | 0.2 | 1 | $\tau'$ | 0.05 | 0.1 | 0.2 | 1 | $\alpha$ | 0 | 0.1 | 0.5 | 1 |
|---|---|---|---|---|---|---|---|---|---|---|---|---|---|---|
| Avg. R@1 | 54.80 | 54.41 | 54.37 | 54.37 | Avg. R@1 | 54.20 | 54.80 | 54.44 | 54.09 | Avg. R@1 | 53.30 | 54.80 | 54.41 | 53.21 |

## C.3. Adaptating Baselines for Missing Modalities

For comparison with baselines under the missing-modality scenario, we adapt these methods with the following minor modifications. For the GRAM (Cicchetti et al., 2025b) and PMRL (Liu et al., 2026) methods, we can still calculate the GRAM matrix for the observed modalities. For instance, if given a V-T dataset, the GRAM matrix **G** should be 2-by-2. In this case, GRAM only needs to minimize one singular value, thus approaching PMRL. However, GRAM sets text as the fixed anchor modality by default, which may limit its optimization. For TRIANGLE (Cicchetti et al., 2025a), it is explicitly designed for datasets with three modalities since it optimizes the area of a triangle spanned by the three modalities. Therefore, for datasets with two modalities, we change it to maximize the cosine similarity between the two related modalities for stable training.

## C.4. Warm-up Training

We employ the warm-up training for initialized generative parameters to avoid the cold-start issue. VAST-150K with complete modalities present is selected. To better mimic the missing modality setting, during the warming-up, we randomly choose one modality as the missing and the others as the observations. We only train the model in one epoch.

## C.5. Hyperparameter Setting

We adopt the AdamW optimizer for training the parameters with $\beta_1 = 0.9$ and $\beta_2 = 0.98$. The learning rate is set to $1 \times 10^{-5}$ and we apply the linear schedule with a warm-up ratio being 0.1. We set the temperature parameters $\tau = 0.05$ and $\tau' = 0.1$ and instance matching weight $\alpha$ to 0.1. All the unimodal representations are transformed with the number of dimensions as 512. The training batch size is set to 64. All the experiments are conducted on the device equipped with $2 \times$NVIDIA H100-80GB GPUs.

## D. Additional Results

**Comparison on more metrics.** In Tables 5 and 6, we report the comprehensive results of multimodal retrieval using the metrics Recall@5 and Recall@10, respectively. A higher top K suggests higher absolute performance and a decreased performance gap. Overall, CalMRL achieves outperforming or comparable results compared to the state-of-the-art methods.

**Comparison against diverse methods.** In Table 7, we replace CalMRL's generative model with alternatives (MVAE (Wu & Goodman, 2018) and MoPoE (Sutter et al., 2021)) while keeping everything else fixed. It directly demonstrates that our method is competitive with other alternatives when operating in the same well-structured representation space. We adapt two representative methods, SMIL (Ma et al., 2021) and Knowledge Bridger (Ke et al., 2025), to operate in our representation space and feed into the PMRL alignment pipeline. CalMRL achieves superior results while being simpler and providing theoretical guarantees. We will include these in the revision.

**Comparison on different hyperparameters.** We report the perfromance of CalMRL *w.r.t.* varing hyperparameters in Table 8. We observe that performance is relatively stable across a reasonable range of $\tau$ and $\tau'$, indicating robustness to their specific settings. $\alpha$ shows a clear pattern: removing it entirely or over-weighting it both degrade performance, while moderate values yield comparable results.

## E. Reproducibility

We provide implementation details, including illustrative algorithm descriptions and flows (see Algorithm 1). The source code will be publicly released for reproducibility.

## F. Limitations

CalMRL enhances multimodal learning by flexibly handling datasets with missing modalities. Despite its effectiveness and design rationale, this core idea is model-architecture agnostic. Therefore, incorporating a more powerful backbone (such as recent advanced vision-text models) could significantly improve its capabilities. However, adopting a new backbone necessitates pre-training to adapt these models, which is computationally and data-collection intensive. To this end, we chose a recognized approach, the emerging complete-modality alignment frameworks (*e.g.*, GRAM and PMRL), to validate the effect of CalMRL.

