# OpenReview forum: "Calibrated Multimodal Representation Learning with Missing Modalities"
_ICML.cc/2026/Conference — ICML 2026 regular_

### Official Review · Reviewer_RpEN · 2026-02-21

**Soundness:** 3
**Presentation:** 3
**Significance:** 3
**Originality:** 2
**Overall Recommendation:** 4
**Confidence:** 3

**Summary:**

This paper dives into the missing modality problem in multi-modal representation learning. It identifies a key challenge called anchor shift, where in the missing modality scenarios, observed modalities are aligned with local anchors. To solve this incomplete alignment challenge, it proposes a calibration-based bi-step algorithm that leverages the inherent multi-modal correlations to impute the missing data in the latent space. Theoretical error bound and convergence analysis are provided. The model is trained on VAST-150K, MSRVTT, AudioCaps, and are evaluated with extensive experiments over 10 benchmarks.

**Compliance With Llm Reviewing Policy:**

Affirmed.

**Final Justification:**

The rebuttal has addressed all of my concerns, so I will maintain my supportive score.

**Key Questions For Authors:**

See weakness

**Limitations:**

Yes

**Strengths And Weaknesses:**

Strengths:

1. The paper explores the missing modality problem, which is a practical challenge in deploying multi-modal learning in real-world scenes. The proposed solution is clean and the teaser figure is insightful.
2. The proposed method has a complete theoretical guarantee.
3. The format of writing, figures, and tables is professional.
4. Besides extensive main results, the analysis studies clearly support the claims in the main text.

Weakness and Questions:

1. This paper is motivated by the "anchoring" modeling of multi-modal representation learning, which is typically implemented by a contrastive approach and aims to learn a shared embedding space. However, there are many other widely used self-supervised learning methods [1-2] and architectures [3] for multi-model learning. A smaller scope like multi-modal alignment might be better for justifying the paper's position.
2. What's the entire model pipeline? Do we need a stage to first train multi-modal encoders? Is the calibration loss combined with a self-supervised learning objective? I can infer based on the experiment setting section that the proposed calibration is like a post-training stage that calibrates the learned representation. Maybe I've missed something, but if so, I guess it's better to use a small paragraph to describe the entire workflow.
3. For the experimental setting, why not directly train on the fully missing-modality datasets (MSRVTT and AudioCaps)? What happens if we remove the warm-up stage—will the performance drop? Also, does the warm-up stage actually lead to a less biased multi-modal alignment, or could it introduce extra bias? In section 4.1, since the computation is sufficient to reproduce baseline results under your current setting, is there a reason why you did not evaluate all methods under the fully missing-modality setting as well?
4. Could you list the difference between the calibration method and other imputation based methods (which also impute latent representations, e.g. [4])?



[1] Yu J, Wang Z, Vasudevan V, et al. Coca: Contrastive captioners are image-text foundation models[J]. arXiv preprint arXiv:2205.01917, 2022.

[2] Dong X, Bao J, Zheng Y, et al. Maskclip: Masked self-distillation advances contrastive language-image pretraining[C]//Proceedings of the IEEE/CVF conference on computer vision and pattern recognition. 2023: 10995-11005.

[3] Wang P, Bai S, Tan S, et al. Qwen2-vl: Enhancing vision-language model's perception of the world at any resolution[J]. arXiv preprint arXiv:2409.12191, 2024.

[4] Zhang C, Chu X, Ma L, et al. M3care: Learning with missing modalities in multimodal healthcare data[C]//Proceedings of the 28th ACM SIGKDD conference on knowledge discovery and data mining. 2022: 2418-2428.

---

> ### Author Rebuttal · Authors · 2026-03-31
>
> We sincerely thank the reviewer for the thoughtful questions. We address each point below.
>
> *W1. Paper scope clarification*: This paper is motivated by the "anchoring" modeling of...justifying the paper's position.
> >We thank the reviewer for this suggestion and largely agree. In fact, we have scoped the problem in the paper: the Introduction defines our target as a multimodal learning goal where modalities are pulled toward a common anchor direction, and Section 2 (Preliminaries) formally defines the learning objective via the GRAM matrix and its singular values. CoCa and MaskCLIP combine contrastive and generative objectives for vision-language pretraining, including the decoder learning; Qwen2-VL is a large-scale multimodal LLM. Their focus is mixed rather than multimodal representation. Our work is more focused and inherently follows previous work, like CLIP, ImageBind, and PMRL. We are happy to add a brief remark distinguishing our target paradigm from these other lines.
>
>
> *W2: Pipeline clarity*: What's the entire model pipeline?...paragraph to describe the entire workflow.
> >The pipeline is described across Sections 3, 4.1, and Appendix C.4, but we agree that consolidating it in one place would improve readability. To directly answer the reviewer's questions:
> >**Yes, a complete-modality warm-up stage is required.** This is not a limitation, but a deliberate and necessary design choice rooted in the problem structure. Without prior exposure to all modalities, the generative parameters for unseen modalities are undefined. There is no representation geometry to impute into. Attempting to impute an unknown modality space is not a missing-modality problem but a  **modality generalization** problem, which is fundamentally harder and orthogonal to our contribution. Empirical studies on cross-modal generalization (e.g., ModalBed) confirm that zero-prior transfer yields near-random performance. For CalMRL learning, as shown in Algorithm 1, since we derive the closed-form solution for updating the generative parameters, we do not need an explicit loss with the self-supervised learning loss. We will add a consolidated pipeline paragraph in Sec 3.2 to enhance the clarity.
>
> *W3: Why not train directly on fully missing-modality data? Any bias from warm-up stage?*. For the experimental setting, why not directly train...missing-modality setting as well?
> >Training directly on fully missing-modality data without any warm-up is not a meaningful baseline for CalMRL. It is a fundamentally different (and much harder) problem. Consider training only on MSRVTT (vision-text): the model has *zero* audio samples. The generative parameters for audio are random, so any imputed audio representation is pure noise. The model is not imputing a missing modality—it is being asked to *invent* an entirely unknown modality space from nothing. This is modality generalization, not the problem we solve. For the bias in the warm-up stage, complete-modality data for warm-up provides the least biased estimate of cross-modal relationships because all pairwise interactions are observed simultaneously. Any single bimodal dataset provides a strictly more biased view, as it can only capture a subset of cross-modal correlations. To this end, evaluating any of these methods without a warm-up would yield meaningless results, so it would not constitute a fair comparison. We argue that our experimental protocol is the correct controlled setup.
>
> *W4: Difference from other imputation-based methods*. Could you list...latent representations, e.g. [4])?
> >We appreciate the comparison. We list the following fundamental differences. 1. M3Care retrieves similar patients from a database. In our setting, there is no retrieval database of representation vectors to index, where the multimodal data size is much larger than the patient dataset. 2. Their imputation is based on the auxiliary information from other patients. CalMRL imputes in an instance-wise manner, relying on the shared information to bridge observed and missing modalities. 3. M3Care uses Graph propagation to adaptively fuse different information, thus imputing the missing one for the healthcare task. However, CalMRL proposes a different way, offering a rigorous derivation from the anchor view to provide a closed-form solution for calibrating the alignment. We have already included this paper in our discussion in Appendix B. We will also highlight that CalMRL's novelty lies not in "doing imputation" but in (1) the anchor shift theory specific to alignment, (2) the closed-form calibration mechanism tightly integrated with the alignment objective, and (3) the theoretical guarantees on when imputation reduces anchor shift.

---

> > ### Author Rebuttal · Reviewer_RpEN · 2026-04-01
> >
> > All concerns have been addressed

---

> > > ### Author Response · Authors · 2026-04-01
> > >
> > > We sincerely thank you for the positive feedback. We are glad that our response regarding the technical pipeline, paper positioning, and experimental depth fully addressed your concerns. We would be grateful if you could re-evaluate the current score. Thank you for your professional support!

---

### Official Review · Reviewer_yo2j · 2026-03-09

**Soundness:** 2
**Presentation:** 2
**Significance:** 3
**Originality:** 3
**Overall Recommendation:** 4
**Confidence:** 4

**Summary:**

To address the issue of missing modality, the authors provide theoretical insights from an anchor shift. Accordingly, the authors propose CalMRL for multimodal representation learning to calibrate incomplete alignments caused by missing modalities. Extensive experiments and comprehensive analyses demonstrate the superiority of CalMRL.

**Compliance With Llm Reviewing Policy:**

Affirmed.

**Key Questions For Authors:**

Please refer to weaknesses.

**Limitations:**

Yes.

**Strengths And Weaknesses:**

Strengths

1.The paper is well-written and well-organized.

2.The proposed method is theoretically supported.

3.The experimental results demonstrate the effectiveness of proposed method.

Weaknesses

1.The symbol notation can be improved. For example, the meaning of $\mathcal{M}$ is missing; The symbol $\tilde{z}$ first appears on the right-hand side in line 107, but it is not explained until line 119; $m \neq m’$ should be emphasized.

2.The selected baselines (PMRL, TRIANGLE, and GRAM) focus on the self-supervised learning involving more than 2 modalities, thereby leading to the lack of comparisons with baseline with respect to missing modalities such as [1,2].

3.From my standpoint, the left singular vectors can only measure direction because all the singular values have been normalized. Therefore, in Theorem 1, it is not reasonable to quantify the anchor shift solely using the leading left singular vector; rather, it should be jointly determined by both its angular deviation and its magnitude.

4.In Equation (2), why do \beta and m denote shared latents and uniqueness? It also seems reasonable that if $\beta$ and $\mu$ denote the uniqueness and the shared latents, respectively.

5.In the proof of convergence analysis, the upper bound of log-likelihood should be derived. Otherwise, the conclusion that the method can converge to a stationary does not hold.

6.Indeed, the introduction of results under the continual setting is confusing. I think more experimental results on benchmarks such as [1, 2] should be provided.

7.As for the t-SNE results, existing works [3,4] have verified that maintaining the modality gap is beneficial for multimodal learning, which contradicts the t-SNE results. Any explanation can be provided ?

---

> ### Author Rebuttal · Authors · 2026-03-31
>
> We sincerely thank the reviewer for the thorough and rigorous review. We address each point below.
>
> *W1. Symbol notation needs improvement*: The symbol notation...$m\neq m'$ should be emphasize
> > Thank you for your careful reading. We will add explicit definitions at first use: $\mathcal{M}$ (full modality set), $\tilde{z}$ (representations of observed modalities), and emphasize $\forall\{m,m'\}\subset\mathcal{M}$ with $m !=m'$.
>
> *W2. Missing baselines from missing-modality literature*. The selected baselines...missing modalities such as [1,2].
> > We appreciate this suggestion. CalMRL targets aligning modal representations and learn a unified embedding space. Prior missing-modality methods are typically designed for task-specific fusion and prediction with specialized architectures, making direct comparison on our benchmarks non-trivial.
> >
> >To address your concern, we have adopted two representative methods to operate within our representation space and alignment pipeline:
> >||MSRVTT|ActivityNet|DiDeMo|VATEX|AudioCaps|ClothoV2|Avg.|
> >|-|-|-|-|-|-|-|-|
> >|**CalMRL (Ours)**|61.1|55.4|57.1|81.3|50.1|23.8|**54.8**|
> >|SMIL|60.2|54.8|55.0|80.6|49.4|24.2|54.0|
> >|Knowledge Bridger|60.9|54.7|55.1|80.8|48.9|24.5|54.2|
> >We will include these results in the revision.
>
> *W3. Anchor shift should consider both angular deviation and magnitude*. From my standpoint, the left singular vectors...its magnitude.
> >Thanks for raising this interesting point, which allows us to clarify a deliberate design choice. The anchor $u_1$ is precisely the consensus direction toward which all modalities are driven. Magnitude shift is secondary and already controlled: since $\|z^m\|=1$, the singular value $\sigma_1$ reflects alignment quality (achieving $\sigma_1=\sqrt{k}$ when perfectly collinear), not an independent degree of freedom. Specifically, $|\sigma_1-\sigma_1^\Omega|\leq\|Z_{\bar{\Omega}}\|_2\leq\sqrt{|\bar{\Omega}|}$ by Weyl's inequality, and this bound does not interact with the spectral gap $\sigma_1-\sigma_2$. The directional bound is already analytically rich because it reveals the interplay between alignment quality and missing modality energy.
>
> *W4. Why does $\beta$ denote shared latent and $\mu^m$ denote modality-specific offset, rather than the reverse?* In Equation (2), why...latents, respectively.
> >The answer reveals a fundamental modeling rationale about cross-modal space heterogeneity. Different modalities inhabit structurally different representation spaces. In our model, the shared latent $\beta$ captures instance-specific semantics in a modality-agnostic coordinate system. $W^m$ projects this into modality $m$'s specific space, while $\mu^m$ absorbs global distributional offset. In the reverse assignment, a shared $\mu$ would be added *directly* to every modality without transformation, implying all representation spaces are already aligned, and the same fixed vector is simultaneously meaningful across modalities, which contradicts the well-known modality gap phenomenon.
>
> *W5. Convergence proof clarification*. In the proof of convergence...does not hold.
> >Thanks for the reviewer's rigor. Corollary 4 establishes that the observed-data log-likelihood is monotonically non-decreasing. The log-likelihood of the linear-Gaussian model is bounded above by $-\frac{N}{2}\log((2\pi)^d\det(V))$. Combined with Corollary 4, convergence to a stationary point follows from the Monotone Convergence Theorem. We will make this argument more explicit in the revision.
>
> *W6. Continual setting presentation*: Indeed, the introduction of results under the continual...should be provided.
> >We appreciate the feedback on presentation clarity. The imputation-calibration method requires a pretrained generative model with learned cross-modal correlations. Without it, imputation becomes a cold-start modality generalization problem. This reflects real practice where alignment models are first trained on complete data, then improved with abundant but incomplete data. Our evaluation already covers 4 vision-text, 2 audio-text, and 4 classification datasets. We are happy to extend to the suggested benchmarks if available.
>
> *W7. t-SNE results that show reduced modality gap*: As for the t-SNE...explanation can be provided?
> > This is an insightful observation, and thanks for raising it. We will address the general argument by discussing the most relevant works on this topic, and then clarify why CalMRL's behavior is not contradictory. Whether the modality gap is beneficial depends on what you're optimizing for. In fusion, the gap preserves complementary information; in alignment/retrieval, the gap is exactly what you're trying to reduce. A large modality gap in this setting means retrieval fails—the system cannot bridge modalities. Even in this setting, we do not claim that the gap should be zero. CalMRL reduces the gap toward what complete-modality training would produce, not beyond it.

---

> > ### Author Rebuttal · Reviewer_yo2j · 2026-04-03
> >
> > All concerns have been addressed.

---

> > > ### Author Response · Authors · 2026-04-03
> > >
> > > We sincerely thank the reviewer for the positive feedback. We are glad that our detailed response, covering the notation refinements, additional empirical comparison, theoretical clarifications, and the modality gap discussion, has fully addressed your concerns. We would be deeply grateful if you could consider re-evaluating the score. Thank you for your time and professional guidance!

---

### Official Review · Reviewer_HVpn · 2026-03-10

**Soundness:** 3
**Presentation:** 2
**Significance:** 3
**Originality:** 3
**Overall Recommendation:** 4
**Confidence:** 5

**Summary:**

This paper proposes CalMRL for multimodal representation learning to calibrate the incomplete alignment caused by missing modalities. The framework leverages prior knowledge and intrinsic correlations between modalities to perform imputation modeling of missing modalities at the representation level, and then adopts a two-step learning approach for optimization.

**Compliance With Llm Reviewing Policy:**

Affirmed.

**Key Questions For Authors:**

(1) The authors should unify the mathematical notation throughout the manuscript and provide explicit definitions for all symbols used.

(2) The authors should refine the writing throughout the manuscript to improve clarity and professional expression.

(3) The authors should clarify why the methodology is designed for multiple modalities, yet the experiments are conducted using only two-modality datasets.

(4) The authors should include additional parameter sensitivity experiments to further validate the robustness of the proposed method.

**Limitations:**

yes

**Strengths And Weaknesses:**

The advantages of this paper are as follows: (1) In the paper, a Calibrated Multimodal Representation Learning (CalMRL) framework is designed. This framework is able to address the overlooked missing modalities dilemma by calibrating incomplete alignment. (2) In the paper, a generative model is proposed to impute missing modalities, thus compensating for anchor shift. The imputation precision is refined by iterating the posterior inference and parameter optimization with theoretical grounding, followed by optimizing encoders by incorporating both observed and imputed modalities.

The paper has the following aspects that can be improved:
(1)	The mathematical notation throughout the paper needs to be unified and refined. For instance, mu_m in Equation (2) on page 3 is inconsistently described as mu_m elsewhere in the text. In Algorithm 1, it remains unclear whether the subscript i in xi_{i=1}^N,m_{i}, and V_{i} denotes the same index or represents different entities. Furthermore, many symbols are introduced without definitions, leading to significant confusion for the reader.

(2)	The manuscript suffers from pervasive linguistic clarity issues. For instance, the statement in the paragraph starting at line 68, '...which leads to anchor shift in our theoretical analysis,' is obscure and difficult for the reader to comprehend. Furthermore, the presentation of Algorithm 1 is non-standard.


(3)	In the experimental section, despite some datasets containing multiple modalities, only two were ultimately selected for the experiments. Is CalMRL restricted solely to bi-modal conversion?

(4)	The manuscript mentions a series of parameters, but lacks a parameter sensitivity analysis.


(5) In Table 5 and Table 6 of the Appendix, the optimal results are not highlighted in bold.

---

> ### Author Rebuttal · Authors · 2026-03-31
>
> We sincerely thank the reviewer for the detailed comments and constructive suggestions. We respond to each point below.
>
> *W1&Q1. Inconsistent notation and undefined symbols*: The mathematical notation throughout...for the reader.
> >Thanks for the careful reading. We will unify all notation in the revision: mu^m consistently denotes modality-specific mean throughout; subscript $i$ uniformly denotes instance index for $x_i$, $m_i$, and $V_i$ in Algorithm 1. All symbols will be formally defined at first use, supplemented with a notation table in the appendix.
>
> *W2&Q2. Writing clarity and non-standard algorithm presentation*: The manuscript...is non-standard.
> > Line 68 will be revised to: "...which we theoretically show leads to anchor shift in the representation space." We use the non-standard algorithm flow to highlight our core implementation. We will revise it to follow a more standard format: Add explicit Input/Output declarations (replacing Require/Ensure); Separate the EM subroutine into a clearly labeled inner block; Add inline comments explaining each step's purpose; Number the bi-step optimization to match the notation used in Section 3. We will also conduct a full pass to address other clarity issues, including tightening informal phrasing and ensuring all technical terms are defined before use.
>
> *W3&Q3: Only bimodal evaluation despite multi-modal design*. In the experimental section,... Is CalMRL restricted solely to bi-modal conversion?
> >Thanks. CalMRL supports arbitrary $k$ modalities: Eq. (2), posterior Eq. (5), and the PMRL objective all handle any $k$. Bimodal training datasets reflect the real-world prevalence of incomplete modality data. Crucially, warm-up on VAST-150K covers 4 modalities (video, audio, caption, and subtitle), and Stage 2 retains two prevailing modalities for each dataset. Therefore, the generative model actively learns to impute in a 4-modal setting. Bimodal evaluation benchmarks reflect the standard retrieval protocol.
>
>
> *W4&Q4: Missing parameter sensitivity analysis*: The manuscript mentions a series of parameters, but lacks a parameter sensitivity analysis.
> >Thanks. The hyperparameters are mainly $\tau$, $\tau'$, and $\alpha$, which are initialized following PMRL's recommended defaults. To address your concern, we have added the following hyperparameter analysis below:
> >|$\tau$|0.05|0.1|0.2|1|
> >|-|-|-|-|-|
> >|Avg. R@1|54.80|54.41|54.37|54.37|
> >
> >|$\tau'$|0.05|0.1|0.2|1|
> >|-|-|-|-|-|
> >|Avg. R@1|54.20|54.80|54.44|54.09|
> >
> >|$\alpha$|0|0.1|0.5|1|
> >|-|-|-|-|-|
> >|Avg. R@1|53.30|54.80|54.41|53.21|
> >
> >Due to rebuttal time constraints, we cannot conduct a full grid search. Nonetheless, we observe that performance is relatively stable across a reasonable range of $\tau$ and $\tau'$, indicating robustness to their specific settings. $\alpha$ shows a clear pattern: removing it entirely or over-weighting it both degrade performance, while moderate values yield comparable results.
>
>
> *W5*: In Table 5 and Table 6 of the Appendix, the optimal results are not highlighted in bold.
> > Thank you. We will bold the best results in the final paper.

---

> > ### Author Rebuttal · Reviewer_HVpn · 2026-04-02
> >
> > Regarding Q3, I find the authors' response to be insufficient. Although the methodology is theoretically capable of handling multiple modalities and the warm-up stage utilized a 4-modal dataset, the subsequent generative experiments were limited. Specifically, the reported results only demonstrate the scenario where one modality is used to impute one other missing modality.
> >
> > To fully validate the claimed multi-modal capabilities, the authors should provide experimental results for more complex scenarios, such as the one illustrated in Figure 1, where two or more modalities are missing simultaneously. Demonstrating performance under these higher-order missing patterns is crucial to support the paper's core claims.

---

> > > ### Author Response · Authors · 2026-04-02
> > >
> > > Thank you for your continued engagement and for the opportunity to clarify our experimental setup. We appreciate your rigor regarding the validation of multi-modal capabilities.
> > >
> > > We would like to take this opportunity to provide a more detailed clarification of our experimental setup, as there may have been some ambiguity in how the modality configurations were presented.
> > > The reviewer noted that results appeared limited to "one-to-one" imputation. However, our subsequent generative experiments are conducted using scenarios with two observed modalities and two missing modalities.
> > >
> > > Specifically, we utilize datasets where two modalities are present (e.g., Vision and Text) to simultaneously impute the remaining two missing modalities (e.g., Audio and Subtitles).
> > > These experiments directly manifest the higher-order missing patterns illustrated in Figure 1. We did not drop one existing modality as the missing modality in these bi-modal datasets.
> > > Rather, we deliberately selected datasets that inherently lack some modalities (2 missing) compared to the full four-modal set.
> > > This reflects the real-world challenge where mainstream datasets often provide an incomplete subset of modalities.
> > >
> > > We believe these existing results support our core claim regarding the model's ability to handle complex, multi-modal missing patterns. We will refine the Sec. 4.1 Experimental Setup to explicitly detail these configurations and avoid any further misunderstanding.
> > >
> > > We hope that the above clarification fully addresses your concerns and provides the necessary confidence in our experimental validation.

---

### Official Review · Reviewer_dYk9 · 2026-03-12

**Soundness:** 3
**Presentation:** 3
**Significance:** 3
**Originality:** 2
**Overall Recommendation:** 3
**Confidence:** 3

**Summary:**

The paper studies multimodal representation learning in missing-modality settings by identifying the anchor shift phenomenon, which occurs when training alignment from incomplete modality sets and the learned consensus direction (anchor) deviates from the ideal one defined by all modalities. The authors propose CalMRL, which uses a shared-latent generative model to impute the missing modality representations in closed form. They propose analytic posterior inference of the shared latent and EM-like closed-form updates for generative parameters, and then performing an alignment objective on the concatenation of observed and imputed embeddings. The paper provides spectral bounds on anchor shift, a sufficient condition under which calibration reduces the shift, and a monotonicity guarantee for the generative subroutine. The experiments on video–text and audio–text benchmarks show improvements.

**Compliance With Llm Reviewing Policy:**

Affirmed.

**Key Questions For Authors:**

Here are my questions for the authors (also please look at the weakness section:

1. It would be better to include the missing related works

2. Multimodal VAEs can handle missing modlaities. How is your method better than [1,2,3, 4] and others?

3. Is there any experiment on increasing portions of missing modality portions ablation study?

[1] Wu, Mike, and Noah Goodman. "Multimodal generative models for scalable weakly-supervised learning." Advances in neural information processing systems 31 (2018).

[2] Wesego, Daniel, and Amirmohammad Rooshenas. "Score-based multimodal autoencoders." TMLR (2024).

[3] Sutter, Thomas M., Imant Daunhawer, and Julia E. Vogt. "Generalized multimodal ELBO." ICLR (2021).

[3] Wesego, Daniel, and Pedram Rooshenas. "Multimodal ELBO with Diffusion Decoders." arXiv preprint arXiv:2408.16883 (2024).

**Limitations:**

yes

**Strengths And Weaknesses:**

Here are some strengths of the paper:

The paper proposes a simple and closed-form representation-level imputation mechanism via a shared latent linear-Gaussian model, with EM-like updates.

Their proposed approach can be compatible with other approaches which can broaden the applicability

Their experimental evaluation looks strong using a wide range of datasets in different modalities with good improvements overall

They support their approach with good theoretical grounding as shown in the paper.

Here are some weaknesses of the paper:

The generative imputation model is linear-Gaussian and shared-latent only; while this enables closed-form solutions, it limits expressivity for complex cross-modal relationships and may underfit the distribution.

Some related works seem missing and not discussed, even though very related, like probabilistic CCA and multi-view factor analysis, group factor analysis, classical EM with missing modalities in multi-view settings, and modern multimodal VAEs (MVAE, SBM-VAE, MoPoE) for missing-modality inference. While some recent works are cited, a more thorough linkage to these foundational lines is needed, given the close methodological connection.

Comparisons are largely against alignment-focused methods adapted to missing-modality training. Other baselines from the missing-modality literature (like multimodal VAEs explicitly handling missing inputs, recent robust missing-modality training methods) are not empirically included

---

> ### Author Rebuttal · Authors · 2026-03-31
>
> We sincerely thank the reviewer for the constructive feedback. We address each point below.
>
> *W1. Linear-Gaussian generative model may lack expressivity*: The generative imputation model is...underfit the distribution.
> >We appreciate this observation. Crucially, the linear-Gaussian model does not operate on raw data. It operates in the **representation space** output by pretrained nonlinear encoders $\phi^m$. These deep encoders handle the heavy lifting of modeling complex cross-modal relationships, projecting raw inputs into a well-structured space. The linear-Gaussian model only needs to capture residual cross-modal correlations among these learned representations, a substantially simpler task. This yields closed-form posteriors, guaranteed convergence (Corollary 4), and a principled imputation error bound (Corollary 3), none of which are available in end-to-end nonlinear alternatives. Empirically, Figure 3 confirms that even this simple generative model can achieve low MSE, and Table 1 shows downstream gains. We view extending CalMRL with lightweight nonlinear imputation as a promising future direction, though our current results suggest the linear-Gaussian model is already a strong and principled choice.
> >We will add this discussion to the revision.
>
> *W2. Missing related works on classical methods and multimodal VAEs*: Some related works seem missing and not discussed...methodological connection.
>
> >Thanks for your suggestion. We will add a dedicated paragraph in the related work to discuss these lines:
> >**Classical methods.** While CalMRL's generative model shares the linear-Gaussian form, we apply it as a lightweight calibration module in the *encoded* representation space, coupled with a modern SVD-based alignment objective. Classical pCCA/GFA operates on raw features, while CalMRL delegates nonlinearity to pretrained encoders and uses the linear model solely for cross-modal imputation. This decomposition enables our anchor-shift analysis (Theorem 1, Corollary 3), specific to the alignment-via-Gramian paradigm used in PMRL/GRAM.
> >**Multimodal VAEs.** MVAE/MoPoE use amortized variational inference with nonlinear encoder-decoder pairs end-to-end. CalMRL instead uses fixed-form closed-form inference, trading decoder expressivity for analytical guarantees and training simplicity. Notably, the PoE structure in our posterior (Eq. 5) is structurally related to MVAE's Gaussian product-of-experts, but with global (not instance-dependent) precision contributions. We will highlight this parallel.
> >
> >We believe these additions will substantially strengthen the paper's scholarly grounding. We will incorporate them in the revision.
>
> *W3&Q2. Missing empirical comparisons with multimodal VAEs and missing-modality methods*. Comparisons are largely against...not empirically included
>
> >We agree that a broader empirical comparison would strengthen the paper. We want to first clarify the scope and then describe the additional experiments we have conducted.
> >
> >**Scope.** CalMRL targets aligning multimodal representation, where the goal is a unified embedding space. Most missing-modality methods are designed for task-specific prediction with specialized fusion modules, and do not produce a general-purpose aligned representation space. Directly comparing retrieval metrics against these methods would involve substantial architectural re-engineering that may not reflect their intended use.
>
> >To address your concerns, we will add the following experiments from two complementary axes:
> >||MSRVTT|ActivityNet|DiDeMo|VATEX|AudioCaps|ClothoV2|Avg.|
> >|-|-|-|-|-|-|-|-|
> >|**CalMRL (Ours)**|61.1|55.4|57.1|81.3|50.1|23.8|**54.8**|
> >|MVAE|61.0|55.0|55.9|80.8|48.4|23.0|54.0|
> >|MoPoE|61.0|55.2|55.6|80.7|48.7|23.5|54.1|
> >|-|
> >|SMIL|60.2|54.8|55.0|80.6|49.4|24.2|54.0|
> >|Knowledge Bridger|60.9|54.7|55.1|80.8|48.9|24.5|54.2|
> >- We replace CalMRL's generative model with alternatives (MVAE & MoPoE) while keeping everything else fixed. It directly demonstrates that our method is competitive with other alternatives when operating in the same well-structured representation space.
> >- We adapt two representative methods, SMIL and Knowledge Bridger, to operate in our representation space and feed into the PMRL alignment pipeline. CalMRL achieves superior results while being simpler and providing theoretical guarantees. We will include these in the revision.
>
> *Q1*: It would be better to include the missing related works.
> >Addressed in W2, and it will be incorporated in the revision.
>
> *Q3*: Is there any experiment on increasing portions of missing modality portions ablation study?
> >Thanks for the question. Table 3 shows results after adding individual missing-modality datasets. Combined with Table 1, the progression, 48.1→51.7→54.2, demonstrates that CalMRL consistently benefits from additional missing-modality data.

---

> > ### Author Rebuttal · Reviewer_dYk9 · 2026-04-03
> >
> > I thank the authors for answering some of my concerns. From the results posted and the similarity with MVAEs, I will keep my score.

---

> > > ### Author Response · Authors · 2026-04-05
> > >
> > > Thanks for your reply. We would like to clarify some distinctions and sincerely hope you will reassess the innovation and contributions of our work. We will explain the distinctions between MVAEs from different perspectives.
> > >
> > > 1) **Conceptually**, CalMRL targets fundamentally different objectives compared to MVAEs. The previously mentioned MVAE-based methods are formulated as density estimation problems, learning $p_\theta(x_1, ..., x_M)$ and performing conditional/unconditional **generation**. The missing modality problem arises naturally; **at test time**, one may need to generate a modality given a subset of observed modalities.
> > > CalMRL focuses on a **representation alignment** problem: learning encoder mappings $\phi:\mathcal{X}\to\mathcal{Z}$ such that instances correspond semantically across modalities. The missing modality problem arises during **training**: directly training the alignment objective on incomplete sets induces a systematic bias, i.e., **anchor shift**. In this case, imputation serves as an **auxiliary subroutine** embedded within CalMRL to help a larger alignment optimization loop. **This distinction fundamentally determines the architecture, optimization,
> > > and evaluation protocol.**
> > >
> > > 2) **Technically**, CalMRL utilizes closed-form solutions to optimize the imputation and performs gradient descent only for the encoder optimization. Whereas, MVAE-based methods adopt **end-to-end gradient-based optimization** of the ELBO. This distinction is basically derived from the different goals and architecture designs in the conceptual discussion above. Imputation occurs in the representation space within our alignment framework.
> > > Furthermore, we contribute **distinct theoretical results**.
> > > CalMRL provides a spectral perturbation analysis of alignment degradation under missing modalities (Theorem 1), derives sufficient conditions for imputation to reduce anchor shift (Corollary 3), and proves monotonic convergence of the EM subroutine (Corollary 4). These results are grounded in matrix perturbation theory and are orthogonal to the ELBO-centric analyses of generative works.
> > >
> > > 3) **Empirically**, these MVAEs evaluate generation quality rather than representation alignment. This is different from CalMRL, which aims to improve unified multimodal representations.
> > > In this setting, the key problem is alignment rather than generation. **A lightweight and resolvable calibration is more desirable.**
> > > Even so, we made efforts to implement these MVAEs in our imputation-calibration framework. The results show that, within our proposed framework, these methods **cannot achieve higher performance than our original design**. This exactly confirms our response to W1, our imputation model is sufficient to calibrate the smooth, low-dimensional representations in multimodal alignment optimization.
> > >
> > > The core similarity with MVAEs, or other classic methods, is the **basic assumption of multimodal learning**, where different modalities are conditionally independent given a shared latent variable. However, we argue that this is a foundational modeling choice that enables tractable inference and principled handling, and is widely adopted by several methods beyond MVAEs. **We follow this recognition and start our analysis of multimodal representation learning with modally incomplete datasets, deriving different theoretical and empirical outcomes than MVAEs.**
> > >
> > > We hope that these explanations can address your concerns, and we would greatly appreciate it if you could consider giving us a higher rating.

---

### Decision · Program_Chairs · 2026-04-30

**Decision:**

Accept (regular)

**Comment:**

All reviewers agree that the paper focuses on a practical problem, i.e. handling missing modalities in multimodal representation learning, and   they also agree that the paper provides good theoretical analysis and empirical results. The authors ran a few additional experiments to address reviewers' concerns. After rebuttal, three reviewers rated above the threshold of weak accept. I recommend weak accept, and I urge the authors to reflect reviewers' suggestions in revision, especially on writing clarity and discussion with related works.